# Effectiveness of aerobic exercise in the prevention and treatment of postpartum depression: Meta-analysis and network meta-analysis

**Hao Xu, Renyi Liu**  **\*, Xiubing Wang, Jiahui Yang**

School of Physical Education, China University of Geosciences (Wuhan), Wuhan, China

\* renyi.liu@foxmail.com

## Abstract

### Background

Aerobic exercise is widely recognized for improving mental health and reducing negative emotions, including anxiety. However, research on its role in preventing and treating post-partum depression (PPD) has yielded inconsistent results. Some studies show positive effects on PPD symptoms, while others find limited impact, suggesting various factors at play, such as exercise type, intensity, and individual differences. To address this gap, our study aims to comprehensively gather evidence on the preventive and therapeutic effects of aerobic exercise for PPD. We'll focus on differences in exercise program design and imple-mentation, exploring how these factors impact intervention outcomes. By identifying effec-tive exercise approaches, we aim to provide more comprehensive exercise prescription recommendations for this vulnerable population.

### Methods

We conducted a quantitative systematic review of the study in 5 representative databases for the effect of aerobic exercise on PPD. Meta-analysis and network meta-analysis were performed with Review-Manager.5.4 and Stata.16.0 software, respectively. This study has been registered on the official Prospero website, and the registration code is CRD42023398221.

### Results

Twenty-six studies with 2,867 participants were eventually included and the efficacy of aero-bic exercise in preventing and treating postpartum depression is significant compared to standard care. (MD = -1.90; 95%CL: -2.58 to -1.21; $I^2$ = 86%). Subgroup analysis suggests that the intervention objective (prevention vs. treatment) of exercise could potentially be a source of heterogeneity in this study, as the "Test for subgroup difference" revealed the presence of significant distinctions (p = 0.02<0.05). The "Test for subgroup difference" yielded non-significant results for both the supervised vs. unsupervised subgroup

**Data Availability Statement:** All relevant data are within the paper and its Supporting Information files.

**Funding:** This work was financially supported by the Fundamental Research Funds for the Central Universities in China (Grant no. CUG150607). The funders did not play a role in the study design, data collection and analysis, decision to publish, or preparation of the manuscript.

**Competing interests:** The authors have declared that no competing interests exist.

comparison (p = 0.55 > 0.05) and the individual vs. team subgroup comparison (p = 0.78 > 0.05). Nonetheless, when assessing their effect sizes [Subtotal (95%CL)], the supervised exercise group [-1.66 (-2.48, -0.85)] exhibited a slightly better performance than the unsupervised exercise group [-1.37 (-1.86, -0.88)], while the team exercise group [-1.43 (-1.94, -0.93)] slightly outperformed the individual exercise group [-1.28 (-2.23, -0.33)]. Network meta-analysis indicated that moderate intensity (35~45 min) group demonstrated a more pronounced intervention effect compared to low intensity (50~60 min) group [-2.63 (-4.05, -1.21)] and high intensity (20~30 min) group [-2.96 (-4.51, -1.41)], while the 3~4 times/week group had a more significant intervention effect compared to 1~2 times/week groups [-2.91 (-3.99, -1.83)] and 5~6 times/week groups [-3.28 (-4.75, -1.81)]. No significant differences were observed in pairwise comparisons of intervention effects among the five common types of aerobic exercises. (95%CL including 0). The Surface Under the Cumulative Ranking curve (SUCRA) results align with the findings mentioned above and will not be reiterated here.

## Conclusion

The efficacy of aerobic exercise in preventing and treating postpartum depression is significant compared to standard care, with a greater emphasis on prevention. The optimal prescribed exercise volume for intervention comprises a frequency of 3~4 exercise sessions per week, moderate intensity (35~45 minutes). Currently, several uncharted internal factors influence the optimal intervention effect of aerobic exercise, such as the potential enhancement brought by team-based and supervised exercise. Given the absence of significant differences in certain results and the limitations of the study, it is essential to exercise caution when interpreting the outcomes. Further research is needed in the future to provide a more comprehensive understanding.

## 1. Introduction

Postpartum depression (PPD) is a common complication following childbirth and is defined as a significant symptom of depression or a typical depressive episode occurring within 1 to 12 months after delivery [1,2]. This condition poses a significant public health threat, affecting not only the physical and mental health of mothers, but also that of their babies [2]. According to 2021 estimates, approximately 13 million women worldwide are diagnosed with PPD each year. Approximately 50% to 75% of mothers encounter mild depressive symptoms, while around 10% to 15% experience postpartum depression within the initial week following childbirth [3,4]. Despite its high incidence rate, the treatment rate for PPD remains low, with 90% of patients going untreated, leading to a substantial burden on families and society as a whole [5]. The traditional treatment for PPD primarily includes psychological and medication interventions. However, the high cost of psychotherapy and the potential side effects associated with antidepressant medications have resulted in low adherence and suboptimal treatment outcomes.

As a new "prescription tool", exercise interventions are not only an important non-pharmacological method in treating postpartum depression, but also effective in preventing this disorder. Aerobic exercise as a common type of exercise for postpartum depression management. Current evidence supports that PPD can be effectively prevented and treated through exercise due to the postpartum-specific health outcomes including less urinary stress incontinence, less lactation-induced bone loss, reducing postpartum weight retention, and less anxiety and

depression [6]. It is widely recognized for the advantages of high practical operability and safety. Costa et al. (2009) conducted a 12-week exercise intervention study and found that aerobic exercise was effective in relieving postpartum depression symptoms in PPD patients [7]. Ren et al. (2019) found a positive effect of aerobic exercise on the treatment of patients with mild to moderate PPD by following up to 12 weeks of intense aerobic exercise in 38 patients with postpartum depression [8]. However, the highly effective method used in aerobic exercise intervention has not yet been fully validated as such. Coll et al. (2019) found that moderate levels of aerobic exercise during pregnancy didn't significantly reduce the patient's postpartum depression symptom scale (EPDS) score [9].

Previous studies have conducted meta-analyses of the efficacy of exercise interventions in PPD prevention and treatment, with aerobic exercise as the primary intervention [10–12]. Moderate exercise can lower the hazard ratio of developing PPD in pregnant women in general [13]. According to evidence, engaging in at least 150 minutes of moderate-intensity aerobic exercise every week can significantly heighten the effectiveness of physical activity in preventing and treating PPD [14]. Moreover, a meta-analysis demonstrated that both low and moderate-intensity exercise can reduce the severity of depressive symptoms among women suffering from PPD [15]. The efficacy of aerobic exercise as an intervention for PPD may be attributed to the combined effects of several factors [16–20], including exercise type, frequency, intensity, duration, supervision, intervention objectives, and whether the exercise was conducted individually or in a group setting. However, at present, there is a dearth of comprehensive data on the most effective aerobic exercise intervention program for preventing and treating PPD. This review study hypothesizes that aerobic exercise may have a positive impact on the prevention and treatment of PPD, but its effectiveness is influenced by various factors, including the purpose of exercise intervention, exercise volume, supervision, exercise mode (group or individual), and individual differences. We will focus on differences in the design of different exercise programs and explore whether these factors affect the effectiveness of aerobic exercise in PPD intervention through both traditional meta-analysis and network meta-analysis. The aim is to provide a more comprehensive and precise exercise guidance for this population, thereby improving their mental health and quality of life.

## 2. Method

### 2.1. Search strategy

A five-step search strategy was conducted (Fig 1) in these domestic and international databases: China National Knowledge Infrastructure (CNKI), Wanfang Database, MEDLINE, Science Direct, PubMed. We included randomized controlled trials (RCTs) that evaluated the prevention and treatment effects of aerobic exercise on postpartum depression in women. The inclusion criteria span from the inception of the database to the present, with studies primarily published in English and Chinese languages and meeting the eligibility criteria for meta-analysis. The following complete search strategy was employed: ((postpartum depression [Title/Abstract] OR postnatal depression [Title/Abstract]) OR (Maternal depression [Title/Abstract] OR Maternal depressive symptoms [Title/Abstract]). The interventions include exercise OR train OR physical activity OR aerobic exercise were selected. See S2 File for specific search strategies.

### 2.2. Inclusion and exclusion criteria

Inclusion criteria: (based on the PICOS principles) (1) The participants are normal pregnant women or postpartum depression patients who are adults (≥18 years). (P: participants); (2) The exercise intervention type in the experimental group was aerobic exercise (I: interventions); (3) Perinatal women who received usual care or other therapies that do not involve

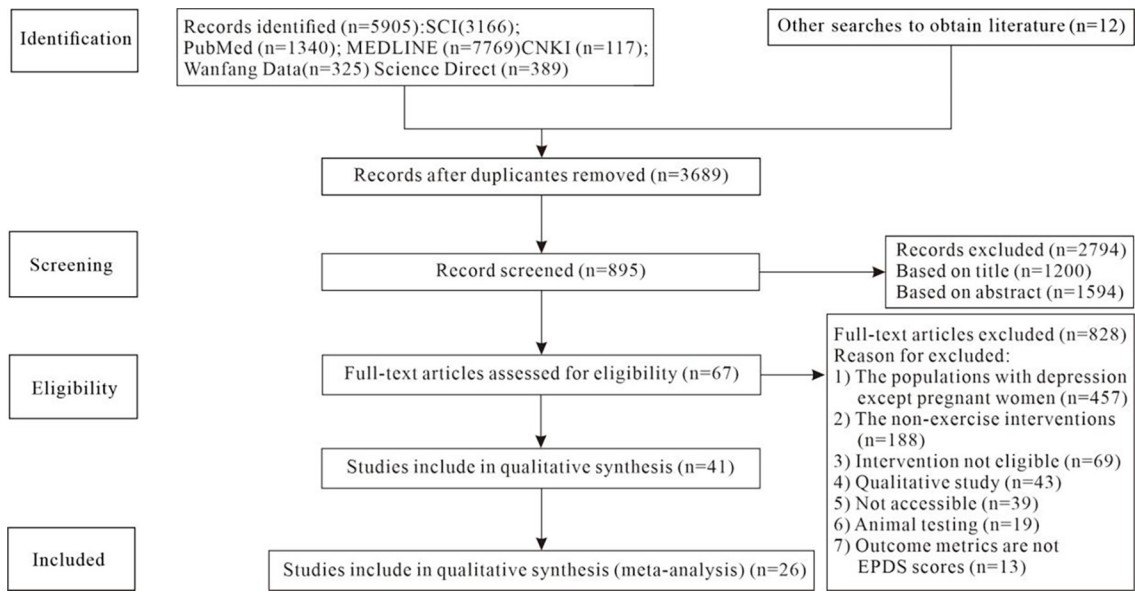

**Fig 1. PRISMA flow diagram of study selection.**

physical activity intervention were as the control group (C: comparisons); (4) The Edinburgh Postnatal Depression Symptom Scale (EPDS) is used to test the severity of postpartum depression symptoms in the subjects (O: outcomes); (5) The analysis type in the literature is a randomized controlled trial (RCT) (S: study design).

Exclusion criteria: (1) The animal testing and the population with depression except for pregnant women (P); (2) No detailed description of the exercise intervention guidelines (I); (3) No control group information (C); (4) Studies in which data are incomplete or valid data cannot be extracted (O); (5) Conference reports, protocols, case reports, reviews, editorial materials, and meta-analyses (S).

## 2.3. Quality assessment and data extraction

According to the preliminary risk assessment guidelines recommended by the Cochrane Collaboration, the following parameters were considered in the analysis: adequate random sequence generation, allocation concealment, blinding of participants and personnel, blinding of outcome assessment, incomplete outcome data, selective reporting and other bias (Fig 2). Two investigators then conducted an independent review of the literature, extracting relevant information and cross-checking to ensure final inclusion. The extracted information included basic details about the studies (author name, publication year), sample characteristics (sample size, location, age), and information about the experimental group intervention (intervention objective, supervision status, type of exercise, intensity, duration, frequency, and total duration of intervention), as well as details about the control group intervention (standard care or other non-exercise interventions), and outcome data (severity of postpartum depression as measured by the Edinburgh Postnatal Depression Scale) (Table 1).

## 2.4. Grouping criteria

The grouping criteria for meta-analysis are typically various variables used in the studies, which can vary based on the specific objectives and questions of the research. In the context of network meta-analysis, grouping criteria can encompass characteristics of different

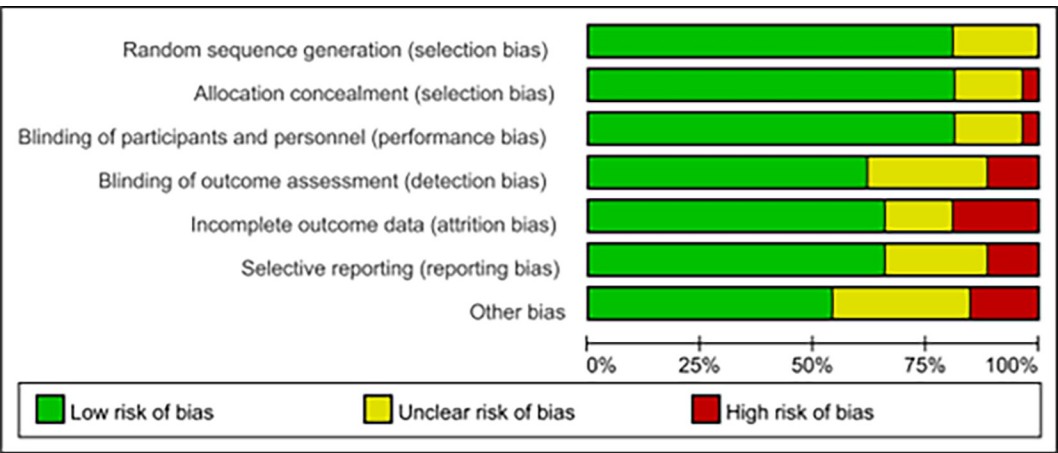

**Fig 2. Bias risk assessment of included studies.**

interventions, treatment plans, or intervention conditions, as well as other relevant factors that might influence intervention effects. To ensure the accuracy of grouping results, apart from the subgroup factors, ensuring the random allocation of other variable factors helps prevent substantial differences in other aspects between the two groups, thereby avoiding any potential impact on the results of subgroup analysis.

Subgroup analysis: After comparing the intervention protocols of the 26 studies, striking differences were found in factors such as the intervention objectives, presence or absence of supervision, and the form of exercise (individual or team). Based on the aforementioned factors, the included studies were subjected to three subgroup analyses. The first subgroup analysis involved categorizing participants according to whether they engaged in exercise individually or with companions during the intervention process. In the same time and space, when only one participant is engaged in exercise, it is classified into the "individual exercise" subgroup. If, in the same time and space, there are other companions besides the participant exercising simultaneously, it is classified into the "team exercise" subgroup. The second subgroup analysis involved grouping participants based on whether the entire exercise intervention was supervised by a fitness expert. Participants who received supervision throughout the exercise intervention were categorized into the "supervised exercise" subgroup, while those without supervision were placed in the "unsupervised exercise" subgroup. The third subgroup analysis was based on the intervention objectives. Studies included in the analysis were categorized into either the "prevention group" if the intervention aimed to prevent PPD, or the "treatment group" if the intervention aimed to treat PPD.

Network meta-analysis: The chosen 26 studies encompass a variety of aerobic exercise types within the experimental group, including cycling/walking/running, yoga, dance, calisthenics, aerobic training classes, swimming, and stretching exercises. Exercise durations range from 20 to 60 minutes, while exercise intensities span high, moderate, and low levels. It's important to provide a clear rationale for grouping cycling/walking/running as a single exercise category, which stems from the common practice in exercise guidelines of combining any two of these activities within training plans. All three of these activities fall under the category of cyclic exercises with relatively low complexity of movement, and their exercise intensities remain consistent. Furthermore, considering the collective weekly exercise volume involving all these aerobic activities, cohorts with similar weekly exercise volumes are categorized into the yoga, dance, and swimming groups. Exercise types that exhibit more diversity, often not limited to a

**Table 1. Information of included studies.**

| Author (year) | Nationality | Age (Mean) (E/C) | Baseline data (EPDS) | Sample size E (n) | Sample size C (n) | Aerobic exercise | Intervention objective (Prevention or treatment) | Exercise form (individual or team) | Supervised (yes or no) | Total duration of intervention (week) | Frequency (time(s)/week) | Intensity | Duration (min) | Control Group | Outcome (EPDS) E (m ± sd) | Outcome (EPDS) C (m ± sd) |
|---|---|---|---|---|---|---|---|---|---|---|---|---|---|---|---|---|
| Yan F (2019) [21] | China | 36.63/36.86 | EPDS≤10 | 101 | 111 | Dance | prevention | TE | no | 8 | 5 | low | 50 | SC | 06.99 ±2.34 | 08.21 ±3.32 |
| Ren Wei (2019) [8] | China | 27.83/28.14 | EPDS≥10 | 19 | 19 | Riding walking | treatment | TE | no | 12 | 3 | low | 50 | SC | 09.94 ±2.32 | 11.42 ±2.03 |
| Li L(2019) [22] | China | 26.71/26.21 | EPDS≥10 | 50 | 50 | Yoga | treatment | IE | no | 8 | 5 | low | 50 | SC | 09.68 ±2.14 | 11.58 ±2.31 |
| Huang L (2003) [23] | China | – | EPDS≤10 | 39 | 31 | Body shape exercise | prevention | TE | yes | 4 | 5 | low | 50 | SC | 07.00 ±4.60 | 05.97 ±5.55 |
| Yang&Chen (2017) [24] | China | 31.89/31.45 | EPDS≤9 | 60 | 62 | aerobic gymnastic | treatment | IE | yes | 12 | 3 | low | 60 | SC | 07.60 ±4.71 | 07.18 ±4.54 |
| Thiruppathi (2014) [25] | India | 26.3/25.1 | EPDS≤9 | 20 | 21 | Aerobic exercise class | prevention | IE | yes | 6 | 5 | high | 45 | SC | 04.95 ±0.68 | 07.52 ±0.51 |
| Surkan (2012) [26] | America | 26.7/26.3 | EPDS≥13 | 203 | 200 | Walking +Stretching | treatment | IE | yes | 18 | 6 | moderate | 35 | SC | 13.30 ±2.76 | 15.30 ±2.73 |
| Shelton (2015) [27] | America | 26.7/25 | EPDS≤9 | 3 | 3 | stroller-walking | treatment | IE | yes | 6 | 3 | low | 60 | SC | 03.00 ±1.00 | 08.00 ±6.00 |
| Saeedi (2013) [28] | Iran | 28.48/27.76 | EPDS≥13 | 20 | 20 | Jogging walking (running) | treatment | IE | yes | 12 | 3 | moderate | 45 | SC | 13.11 ±0.81 | 17.74 ±1.21 |
| Robichaud (2009) [29] | America | 31.1/30.4 | EPDS≥13 | 25 | 23 | walking | treatment | IE | yes | 6 | 3 | low | 30 | SC | 18.08 ±3.28 | 18.39 ±3.68 |
| Teychenne (2021) [1] | Australia | 27.4/27.8 | EPDS≥10 | 25 | 23 | Walking cycling | treatment | – | yes | 12 | 5 | low | 55 | SC | 12.40 ±6.70 | 16.80 ±3.40. |
| Özkan (2020) [30] | Turkey | 28.9/28.63 | EPDS≥10 | 40 | 40 | Dance | treatment | – | no | 4 | 4 | moderate | 45 | SC | 07.29 ±1.67 | 12.54 ±2.65 |
| Norman (2010) [31] | Australia | 29.3/30.1 | EPDS≤9 | 62 | 73 | Aerobic exercise class | prevention | TE | no | 12 | 5 | low | 60 | SC | 05.47 ±5.11 | 06.75 ±5.11 |
| Mohammadi (2015) [32] | Iran | 25.2/25.3 | EPDS≤15 | 38 | 36 | Stretching +Breathing exercise | prevention | IE | yes | 8 | 3 | low | 20~30 | SC | 06.58 ±4.63 | 06.5 ±5.12 |
| Lewis (2018) [33] | America | 31.03/29.77 | EPDS≤9 | 61 | 63 | Aerobic exercise class | prevention | TE | yes | 24 | 5 | moderate | 35 | SC | 04.69 ±3.89 | 07.02 ±4.64 |
| Keller (2014) [34] | British | 28.4/28.4 | EPDS≤9 | 39 | 54 | Yoga | prevention | TE | yes | 24 | 5 | high | 20~30 | SC | 07.05 ±5.36 | 07.80 ±5.05 |
| Heh (2008) [35] | China | – | EPDS≥6 | 63 | 63 | Stretching | prevention | TE | no | 8 | 4 | moderate | 45 | SC | 10.20 ±3.60 | 12.70 ±3.90 |
| Haruna (2013) [36] | Japan | 33.8/33.7 | EPDS≤9 | 48 | 47 | exercise ball | prevention | TE | yes | 12 | 4 | high | 50–60 | SC | 03.60 ±4.20 | 04.10 ±3.40 |
| Forsyth (2017) [37] | British | 25/27 | EPDS≥12 | 11 | 11 | pram-walking | treatment | IE | no | 12 | 2 | high | 20 | SC | 11.80 ±6.10 | 12.70 ±4.20 |
| Daley (2015) [38] | British | 31.7/29.3 | EPDS≥13 | 47 | 47 | Aerobic exercise class | treatment | IE | no | 24 | 1 | high | 30 | SC | 12.51 ±5.46 | 14.67 ±4.86 |
| Daley (2008) [39] | British | – | EPDS≥12 | 16 | 15 | Dance | treatment | IE | no | 14 | 1 | high | 30 | SC | 13.10 ±5.20 | 14.30 ±5.40 |

*(Continued)*

**Table 1.** (Continued)

| Basic Bibliographic Information | | Sample Information | | | | Aerobic exercise | Exercise Guidelines (Experimental Group) | | | | | | | Control Group | Outcome (EPDS) | |
|---|---|---|---|---|---|---|---|---|---|---|---|---|---|---|---|---|
| Author (year) | Nationality | Age (Mean) | Baseline data | Sample size | | | Intervention objective (Prevention or treatment) | Exercise form (individual or team) | Supervised (yes or no) | Total duration of intervention (week) | Frequency (time(s)/week) | Intensity | Duration (min) | | E | C |
| | | (E/C) | (EPDS) | E (n) | C (n) | | | | | | | | | | (m ± sd) | (m ± sd) |
| Costa (2009) [7] | Canada | 34.3/34.7 | EPDS≥13 | 46 | 42 | Yoga | treatment | IE | no | 12 | 2 | high | 20 | SC | 08.60 ±4.71 | 09.00 ±5.61 |
| Coll (2019) [9] | Brazil | 27.2/27.3 | EPDS≥12 | 192 | 387 | Jogging (running) | prevention | TE | yes | 16 | 2 | high | 30 | SC | 04.80 ±3.70 | 05.40 ±4.10 |
| Buttner (2015) [40] | America | 29.81/32.45 | EPDS≤9 | 27 | 29 | Yoga | treatment | IE | yes | 4 | 5 | low | 30 | SC | 05.87 ±6.03 | 08.52 ±5.43 |
| Armstrong (2004) [41] | Australia | – | EPDS≥12 | 9 | 10 | Stroller walking | treatment | TE | yes | 12 | 3 | moderate | 45 | SC | 06.33 ±3.67 | 13.33 ±7.66 |
| Aguilar-Cordero (2018) [42] | Spain | 34.52/33.67 | – | 70 | 70 | Swimming +Riding | prevention | IE | no | 18 | 3 | moderate | 45 | SC | 06.41 ±3.68 | 10.17 ±2.38 |

**NOTE:** EPDS baseline data typically refers to the basic data collected using the Edinburgh Postnatal Depression Scale tool to assess the severity of postpartum depression symptoms. An EPDS score of ≤9 signifies mild postpartum depression symptoms in the subject, and a score of ≥13 suggests significant postpartum depression symptoms. "—" indicates that information is not available here; "m" indicates mean "sd" indicates standard deviation; Exercise intensity was expressed as reserve heart rate (HRR) = (maximal heart rate—resting heart rate) × percentage of intensity + resting heart rate; maximal heart rate = 220-age, low exercise intensity: 40% HRR; moderate exercise intensity: 50%~ 60% HRR; high exercise intensity: 65% ~ 74% HRR; IE: Individual exercise, TE: Team exercise, SC: Standard care, C: Control group, E: Experimental.

singular form but maintain relatively consistent weekly exercise volumes, are assigned to other exercise groups. Exercise volume is organized into three tiers, descending from high to low volume. In terms of exercise frequency, they are classified into three brackets: 1~2 times per week, 3~4 times per week, and 5~6 times per week. Importantly, after closely examining the planned exercise intensities and durations across all included studies, a discernible pattern emerges where higher exercise intensities are frequently coupled with shorter exercise durations. Aligning akin patterns of exercise intensity and duration results in three classifications: high intensity (20~30 minutes), moderate intensity (35~45 minutes), and low intensity (50~60 minutes).

## 2.5. Statistical analyses

Firstly, a meta-analysis was conducted using RevMan 5.3 software on the mean and standard deviation of the Edinburgh Postnatal Depression Scale (EPDS) in the experimental and control groups after the aerobic exercise intervention. Based on this data, we conducted Meta-analysis to calculate the effect sizes (MD) and 95% confidence intervals for the experimental and control groups in each study. By aggregating all these results, we can determine whether exercise intervention is effective in preventing and treating postpartum depression. This determination relies on whether there is a significant difference in the outcomes between the experimental and control groups in the Meta-analysis. In a forest plot within a meta-analysis, MD is used to represent the statistical measure of mean difference between different study groups. It assists us in comprehensively assessing effect sizes and significance across studies in a meta-analysis. If the $I^2 \leq 50\%$ or $p > 0.05$, indicating low heterogeneity, a fixed-effects model was applied. Conversely, if $I^2 > 50\%$ or $p \leq 0.05$, indicating high heterogeneity, a random-effects model was used, and we should conduct subgroup analysis to identify the cause of this heterogeneity. The results of subgroup analysis are comprehensively interpreted using the "Test for subgroup difference", as well as the "Subtotal (95% CL)" for each subgroup.

The network meta-analysis was conducted using Stata 16.0 software, and an evidence network diagram was generated. Since there was no closed loop in the evidence network diagram, no inconsistency test was necessary, and direct comparisons were made. The results were presented in a league table, where the data represents the mean difference (MD) values and 95% confidence interval (CI) values for direct comparisons between different interventions. If MD<0, it means the "column" intervention was superior to the "row" intervention, and vice versa. If the 95% CI did not include 0, it indicated statistical significance ($p < 0.05$), and if it did include 0, the opposite was true. The Surface Under the Cumulative Ranking (SUCRA) for each intervention was calculated, with higher values indicating a better intervention effect. Finally, a funnel plot was also generated, where large sample studies with high precision and low numbers are located at the top and cluster near the center of the combined effect size, while small sample studies with low precision and high numbers are located at the bottom and are symmetrically distributed to the left and right. Sensitivity analysis is a valuable tool for evaluating the robustness and reliability of study findings.

## 3. Results

### 3.1. Search results

Twenty-six articles were selected (Fig 1), 2867 cases were obtained, the publication years ranged from 2003 to 2021 [1,7–9,21–42]. The sample subjects of the 26 RCT studies covered 11 countries and regions. All the RCTs experimental group interventions were aerobic exercise with different contents. The control groups had no exercise interventions, but received

standard care (SC). Table 1 shows the basic information about the included studies (sample information, experimental group information, control group information and outcome data).

## 3.2. Risk of bias

According to the preliminary risk assessment for publication bias as recommended by the Cochrane Collaboration. The overall risk of bias in the 26 included studies were judged to have a low risk of bias (70% low risk, 20% unknown risk, 10% high risk) (Fig 2).

## 3.3. Outcomes of meta-analysis

**3.3.1. The overall intervention effect of aerobic exercise on symptoms of PPD.** In Fig 3, each row of colored circles on the right side represents the 7 risk factors (A~G) of risk bias. The red, yellow, green correspond to high risk, unknown risk, and low risk respectively. Meta-analysis summary results (26 RCTs; MD = -1.90, 95% CL: -2.58, -1.21; $I^2$ = 86%) indicate that aerobic exercise is significantly effective in preventing and treating postpartum depression when compared to the control group with standard care. The mean difference (MD) of -1.90 suggests that the symptom scores for postpartum depression are significantly lower in the experimental group compared to the control group, indicating substantial improvement. The 95% confidence interval ranging from -2.58 to -1.21 indicates that, with 95% confidence, the true mean difference is likely within this range, which doesn't include 0. This further supports the significant effect of aerobic exercise. The heterogeneity analysis results show the $I^2$ = 86%, indicating high heterogeneity that needed the subgroup analysis to find the source of heterogeneity.

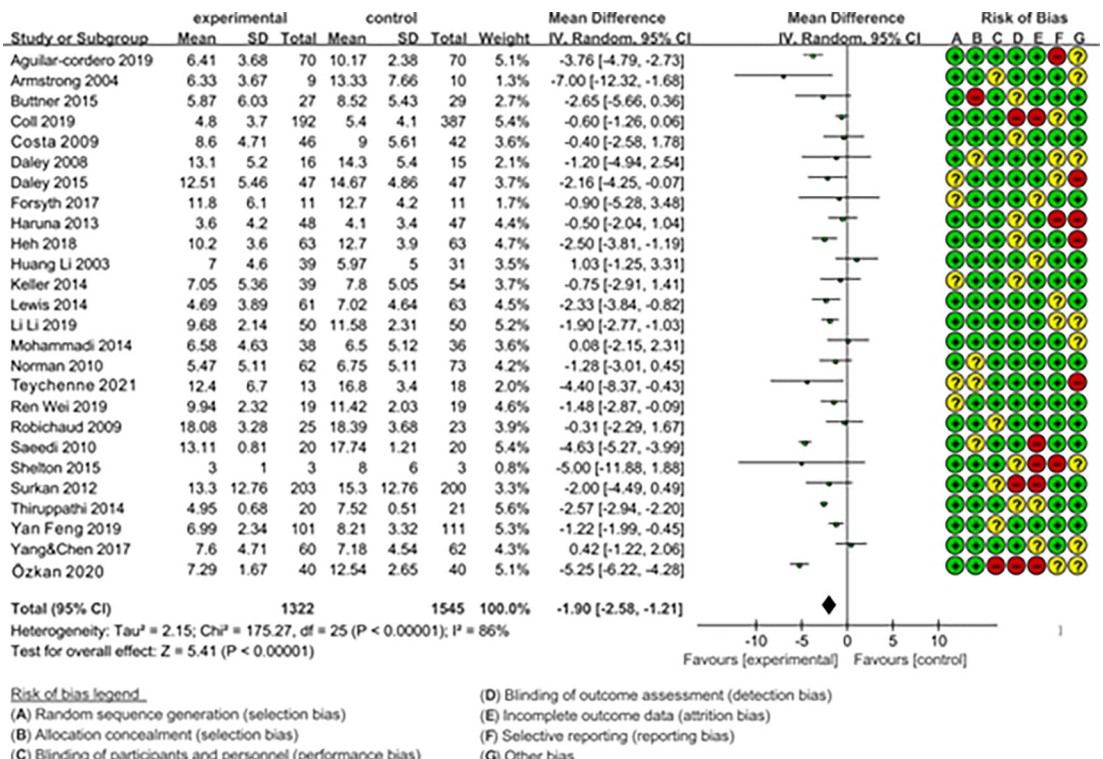

**Fig 3. Forest plot of the overall intervention effect of aerobic exercise on PPD symptoms.**

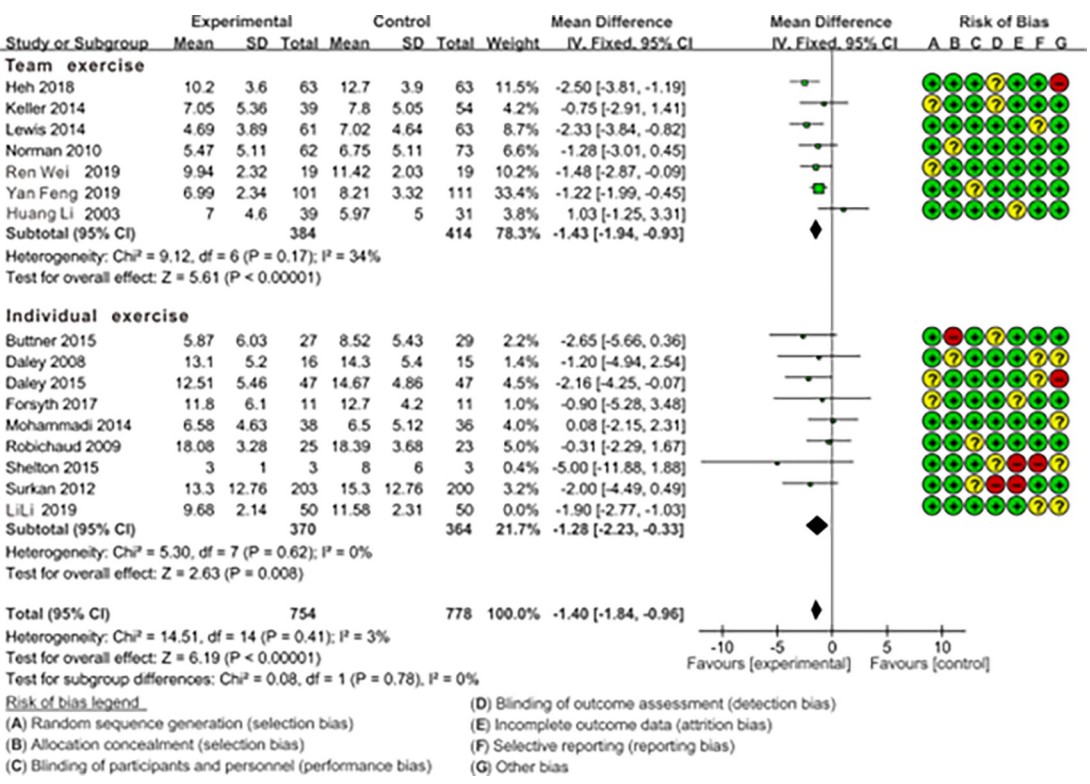

**Fig 4. Forest plot of the comparison of the effect of team exercise vs. individual exercise on PPD symptoms.**

**3.3.2. The impact of individual vs. team exercise on preventing and treating PPD symptoms through subgroup analysis.** With the premise of ensuring the random allocation of other variable factors apart from the subgroup factors, the results of the subgroup analysis are as follows. As shown in Fig 4, Test for subgroup difference indicates no statistical significance (p = 0.78>0.05), indicating that this organizational format of exercise is not the source of heterogeneity in this study. Therefore, both the team exercise group [8,21,31,33–36] (7 RCTs; MD = -1.43, 95%CL: -1.94, -0.93; $I^2$ = 34%) and the individual exercise group [1,7,24,27,37,39,40] (8 RCTs; MD = -1.28, 95%CL: -2.23, -0.33; $I^2$ = 3%), when compared to the control group with standard care are beneficial for reducing postpartum depressive symptoms. Expanding on this, through a direct comparison of the effect sizes (MD Subtotal) between the two groups, it becomes evident that the team exercise group (MD = -1.43) slightly surpassed the individual exercise group (MD = -1.28) in terms of efficacy.

**3.3.3. The impact of supervised vs. unsupervised exercise on preventing and treating PPD symptoms through subgroup analysis.** With the premise of ensuring the random allocation of other variable factors apart from the subgroup factors, the results of the subgroup analysis are as follows. As shown in Fig 5, test for subgroup difference indicates no statistical significance (p = 0.55>0.05), indicating that the supervision of exercise process or not is not the source of heterogeneity in this study. Therefore, both the supervised exercise group [1,24,26,27,29,32,33,40,41] (9 RCTs; MD = -1.66; 95%CL: -2.48, -0.85; $I^2$ = 37%) and the unsupervised exercise group [7,8,21,31,35–39] (9 RCTs; MD = -1.37; 95%CL: -1.86, -0.88; $I^2$ = 9%), when compared to the control group with standard care are beneficial for reducing postpartum depressive symptoms. Building on this, by directly comparing the effect sizes of the two

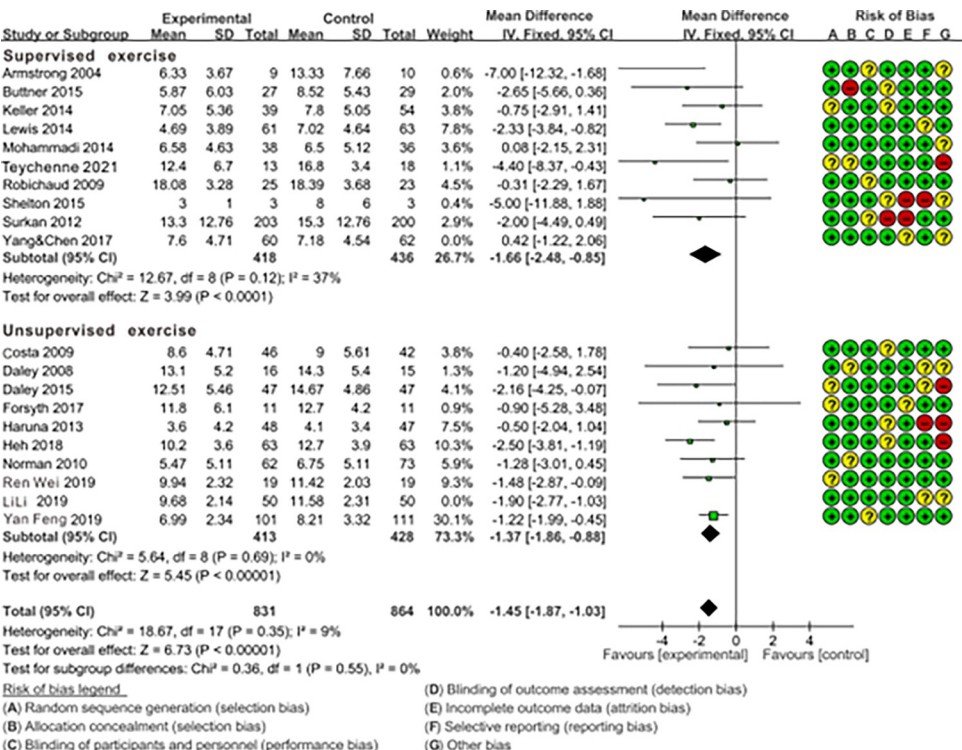

**Fig 5. Forest plot of the comparison of the effect of supervised exercise vs. unsupervised exercise on PPD symptoms.**

groups, it is evident that the supervised exercise group (MD = -1.66) slightly outperformed the unsupervised exercise group (MD = -1.37).

**3.3.4 The impact of the prevention group vs. the treatment group on alleviating PPD symptoms through subgroup analysis.** With the premise of ensuring the random allocation of other variable factors apart from the subgroup factors, the results of the subgroup analysis are as follows. As shown in Fig 6, Test for subgroup difference indicates the presence of significant differences (p = 0.02<0.05), indicating that the intervention objectives may be the source of heterogeneity in this study. And the prevention group (MD = -1.96) is significantly higher than the treatment group (MD = -1.04). Therefore, the prevention group [9,21,23,25,31,32,34–36,42] (10 RCTs; MD = -1.96; 95%CL: -2.23, -1.70; $I^2$ = 84%) was found to be more beneficial than the treatment group [1,7,8,24,26,27,29,37,39,40] (10 RCTs; MD = -1.04; 95%CL: -1.78, -0.30; $I^2$ = 9%) for improving symptoms of PPD. But the heterogeneity in prevention group still remains high ($I^2$ = 84%), indicating the source of heterogeneity in this group is yet to be explored.

## 3.4. Outcomes of network meta-analysis

**3.4.1. The effect of different aerobic exercise programs on improving symptom of PPD.** Evidence network diagram: We employed a network meta-analysis to investigate the impact of different exercise frequencies on the intervention effect. 25 studies were included, and the experimental group was mainly included cycling/walking/running [8,9,26,41], dance group [21,30,39], yoga group [7,22,24,29,34], swimming group [27,36,37,40,42], other sports group [1,23,25,31–33,35,38]. The control group with standard care had no exercise intervention. The network relationship between different exercise content on improving PPD symptoms was shown in Fig 7A.

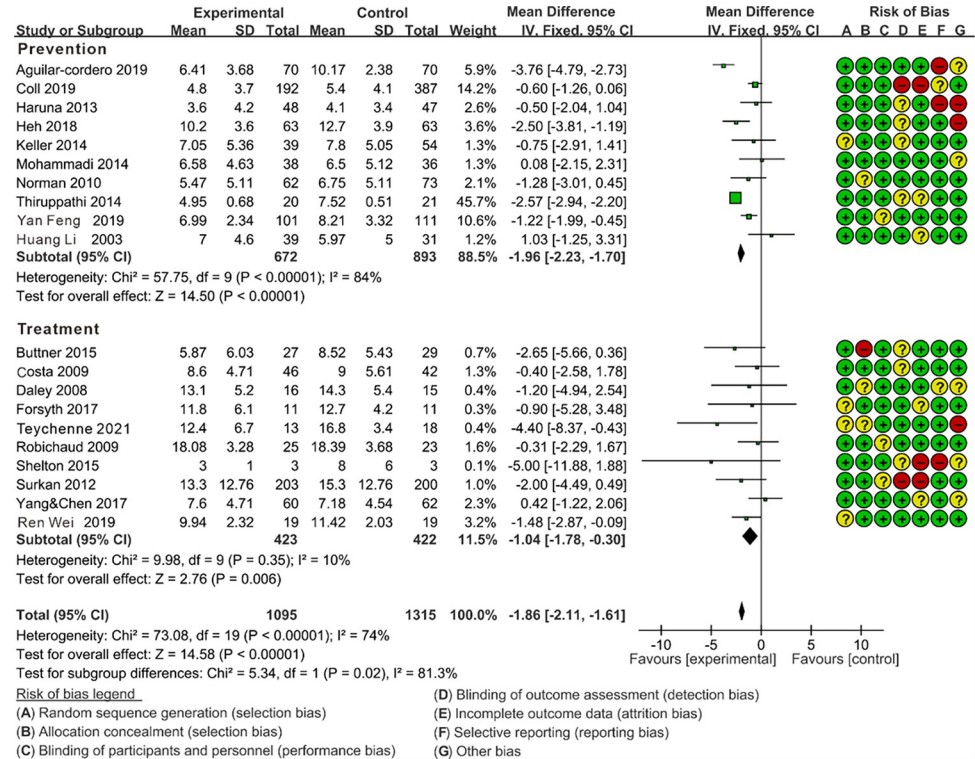

**Fig 6. Forest plot of the comparison of the effect of prevention vs. treatment on PPD symptoms.**

Network meta-analysis: Out of the 15 comparisons, 4 comparisons were found to have a statistically significant difference ($p<0.05$, 95%CL excluding 0). Excluding the yoga group (95%CL including 0), all remaining groups exhibit more favorable intervention effects relative to the control group with standard care. However, pairwise comparisons between different exercise types yield statistically insignificant results (95%CL including 0). (Table 2). SUCRA: dance group (SUCRA = 86.9%) > swimming group (SUCRA = 73.3%) > other sports group (SUCRA = 56.6%) > cycling/walking/running group (SUCRA = 54.4%) > yoga group (SUCRA = 24.3%) > control group (SUCRA = 4.2%) (Fig 7B). Therefore, we cannot draw a conclusive determination regarding which specific exercise type yields superior results in exercise intervention for PPD.

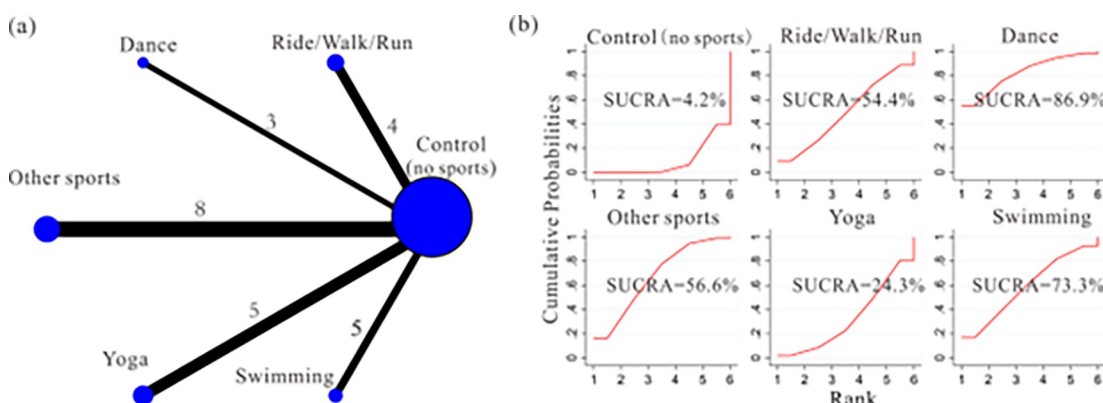

**Fig 7. Network plot and SUCRA represent the effect of different training contents on improving PPD symptoms.**

**Table 2. The effect of different aerobic exercise type on improving PPD symptoms [MD (95% CL)].**

| Aerobic exercise type | Dance | Swimming | Other sports | Ride/Walk/Run | Yoga | Control |
|---|---|---|---|---|---|---|
| Dance | 0 | | | | | |
| Swimming | -0.57 (-2.96,1.83) | 0 | | | | |
| Other sports | -1.15 (-3.21,0.91) | -0.58 (-2.56,1.40) | 0 | | | |
| Ride/Walk/Run | -1.22 (-3.64,1.20) | -0.65 (-2.99,1.68) | -0.07 (-2.07,1.93) | 0 | | |
| Yoga | -2.22 (-4.44, -0.00) * | -1.65 (-3.79,0.49) | -1.07 (-2.84,0.70) | -1.00 (-3.16,1.16) | 0 | |
| Control | -2.89 (-4.63, -1.15) * | -2.32 (-3.96, -0.68) * | -1.74 (-2.85, -0.63) * | -1.67 (-3.33, -0.01) * | -0.67 (-2.05,0.71) | 0 |

NOTE

* indicates $p < 0.05$ (statistically significant difference), because 95% CL of the combined effect size of the measures, with no statistical significance when the 95% CL don't includ 0; When MD<0, indicating that "column" treatment measures were superior to "row" and vice versa; The control group received standard care without exercise intervention.

**3.4.2. The effect of different prescribed exercise volume on improving symptoms of PPD.** *a) Prescribed frequency.* Evidence network diagram: We employed a network meta-analysis to investigate the impact of different exercise frequencies on the intervention effect. 24 studies were included, and the experimental group was mainly included 3 different exercise frequencies, the 1~2 times/week group [7,9,37–39], the 3~4 times/week group [28,30,35,41,42], as well as the 5~6 times/week group [1,8,21–27,29,31–34,36]. The network relationship between different prescribed exercise frequencies on improving symptoms of PPD was shown in Fig 8A.

Network meta-analysis: Out of the 6 comparisons, 4 comparisons were found to have a statistically significant difference ($p<0.05$, 95%CL excluding 0). Among them, a significant difference between the 3~4 times/week and 1~2 times/week groups [-2.91 (-3.99, -1.83)], as well as between the moderate 3~4 times/week and 5~6 times/week groups [-3.28 (-4.75, -1.81)]. (Table 3). SUCRA: The 3~4 times/week group (SUCRA = 100%) > the 5~6 times/week group (SUCRA = 56.8%) > the 1~2 times/week group (SUCRA = 41.3%) > control group

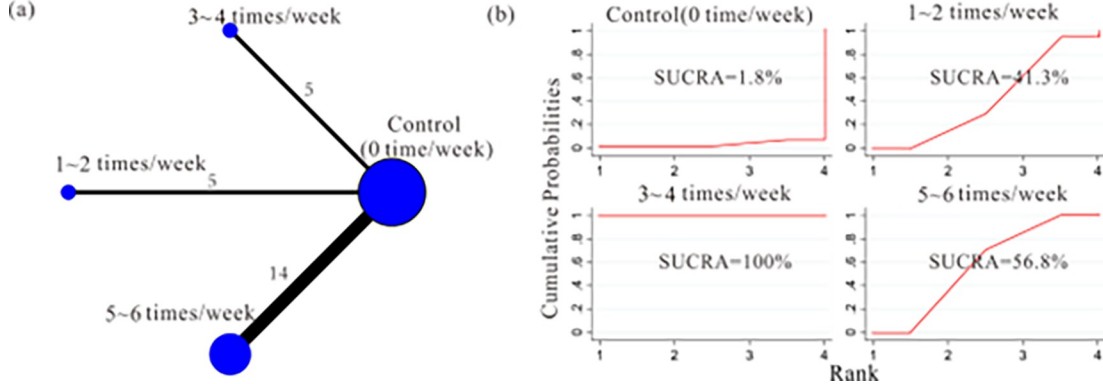

**Fig 8. Network plots and surface under cumulative ranking curves (SUCRA) represent the effect of different prescribed exercise frequency on improving PPD symptoms.**

**Table 3. The effect of different prescribed exercise frequency on improving PPD symptoms [MD (95% CL)].**

| Frequency | 3~4 time/week | 1~2 times/week | 5~6 times/week | Control |
|---|---|---|---|---|
| 3~4 time/week | 0 | | | |
| 1~2 times/week | -2.91 (-3.99, -1.83) * | 0 | | |
| 5~6 times/week | -3.28 (-4.75, -1.81) * | -0.37 (-1.68,0.95) | 0 | |
| Control (0 time/week) | -4.22 (-5.12, -3.31) * | -1.31 (-1.91, -0.70) * | -0.94 (-2.10,0.22) | 0 |

NOTE

* Indicates $p < 0.05$ (statistically significant difference); The control group received routine care without exercise intervention.

(SUCRA = 1.8%) (Fig 8B). Therefore, 3~4 times/week was the best prescribed exercise frequency to improve symptoms of PPD.

*b) Prescribed intensity-duration combinations*. Evidence network diagram: We employed a network meta-analysis to investigate the impact of different prescribed intensity-duration combinations on the intervention effect.16 studies were included, and the experimental group was mainly included 3 intensity-duration combinations. The low (50~60 min) group [1,8,22,27,29], the moderate (35~45 min) group [28,30,33,35,41,42], the high (20~30 min) group [7,9,37–39]. The network relationship between different prescribed exercise intensities-duration on improving PPD symptoms was shown in Fig 9A.

Network meta-analysis: Out of the 6 comparisons, 4 comparisons were found to have a statistically significant difference ($p<0.05$, 95%CL excluding 0). Among them, a significant difference between the moderate (35~45 min) and low (50~60 min) groups [-2.63 (-4.05, -1.21)], as well as between the moderate (35~45 min) and high (20~30 min) groups [-2.96 (-4.51, -1.41)]. (Table 4). SUCRA: The moderate (35~45 min) group (SUCRA = 100%) > low (50~60 min) group (SUCRA = 55%) > the high (20~30 min) group (SUCRA = 42.5%) > control group (SUCRA = 2.5%) (Fig 9B). Therefore, moderate intensity (35~45 min) min was the best prescribed exercise intensity-duration combinations to improve symptoms of PPD.

### 3.5. Sensitivity analysis

The results of the sensitivity analysis demonstrated that the exclusion of each study had minimal impact on the overall findings, underscoring the high level of robustness and reliability of

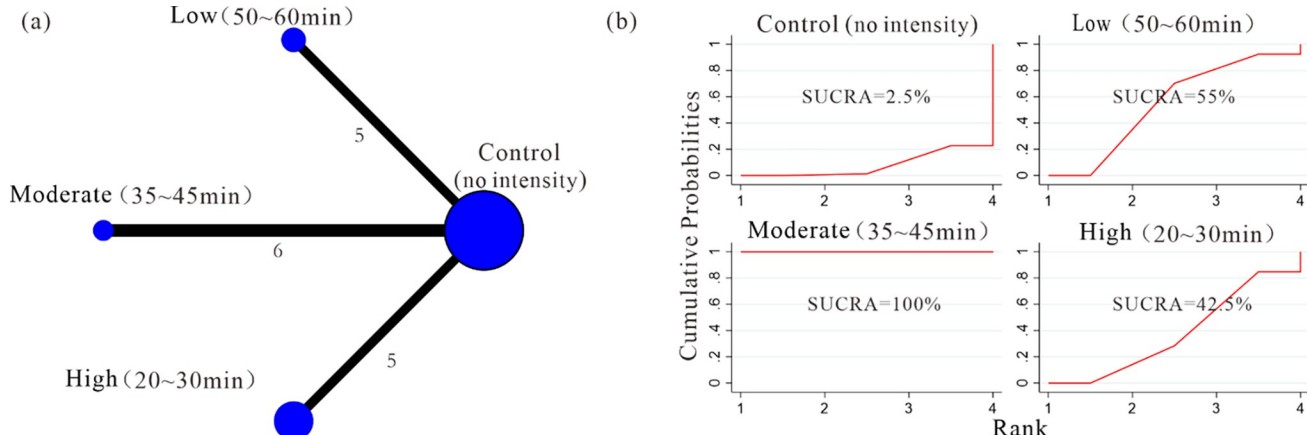

**Fig 9. Network plots and surface under cumulative ranking curves (SUCRA) represent the effect of different prescribed exercise intensity-duration combinations on improving PPD symptoms.**

**Table 4. The effect of different prescribed exercise intensity-duration on improving PPD symptoms [MD (95% CL)].**

| Intensity (duration) | Moderate (35~45min) | Low (50~60min) | High (20~30min) | Control |
|---|---|---|---|---|
| Moderate (35~45min) | 0 | | | |
| Low (50~60min) | -2.63 (-4.05, -1.21) * | 0 | | |
| High (15~30min) | -2.96 (-4.51, -1.41) * | -0.33 (-1.98,1.32) | 0 | |
| Control | -3.92 (-4.83, -3.01) * | -1.29 (-2.38, -0.20) * | -0.96 (-2.20,0.29) | 0 |

NOTE

* Indicates 95%CL excluding 0, $p < 0.05$ (statistically significant difference); low intensity:40%HRR, moderate intensity:50~60%HRR, high intensity:65%~74%HRR; Exercise intensity was expressed as reserve heart rate (HRR) = (maximal heart rate—resting heart rate) × percentage of intensity + resting heart rate, maximal heart rate = 220-age; The control group received routine care without exercise intervention.

this study. The sensitivity analysis influence plot (Fig 10) revealed that 26 studies had a negligible effect on the summary effect size, with the estimated effect size of each study falling within the horizontal line area of the confidence interval.

## 3.6. Publication bias analysis

The symmetrical shape of the funnel plot displayed in Fig 11. suggested that the risk of publication bias was low. In addition, the P-value of Egger's linear regression test, used to evaluate the asymmetry of the funnel plot, indicated the absence of publication bias ($p = 0.32 > 0.05$), as outlined in Table 5.

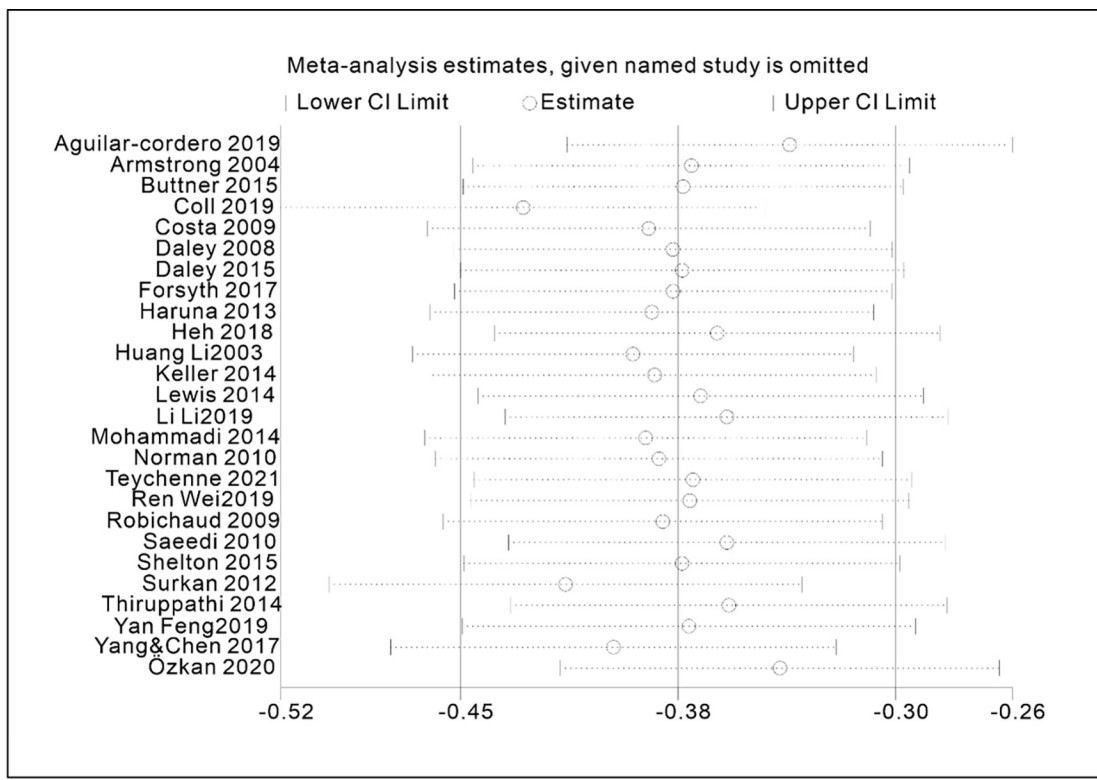

**Fig 10. Sensitivity analysis influence plot of the included studies.**

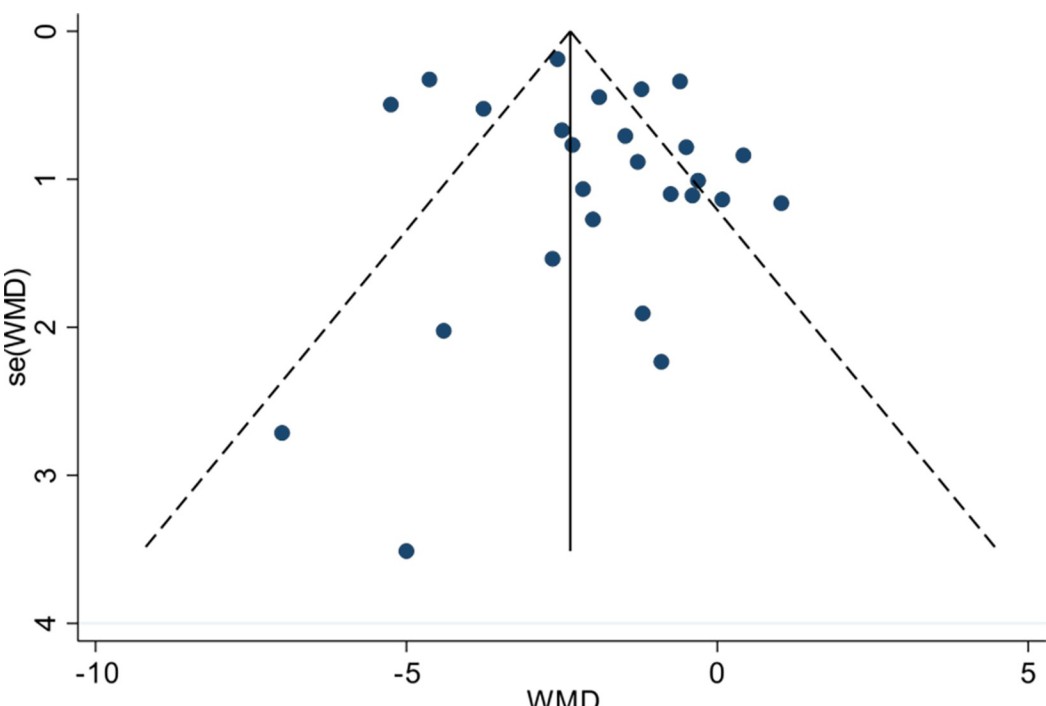

**Fig 11. Funnel plot of publication bias of included studies. NOTE:** The number of dots represents the number of included studies, and its more symmetrical distribution indicates no publication bias, but the description of the symmetry of the distribution is somewhat subjective.

## 4. Discussion

The main results of the meta-analysis show that compared to the control group with standard care, the experimental group engaging in aerobic exercise is more beneficial for preventing and treating postpartum depression. However, it is noteworthy that the study outcomes exhibit a marked level of heterogeneity, suggestive of potential considerable disparities in the effects of interventions among the encompassed investigations. Commencing with the inherent aspects embedded within the exercise intervention guidelines of the experimental group, this study posits that the potency of exercise intervention may be influenced by elements such as the aspirations of the intervention, the organizational format of exercise, the presence or absence of supervision, and the amount of exercise.

Based on the aforementioned hypotheses, we conducted three subgroup analyses. Subgroup analysis suggests that the intervention objective (prevention vs. treatment) of exercise could potentially be a source of heterogeneity in this study (p = 0.02<0.05), indicating that the preventive effects of aerobic exercise are superior to the therapeutic effects. It is well known that engaging in appropriate aerobic exercise during pregnancy not only promotes pelvic mobility

**Table 5. Egger test for publication bias of included studies.**

| Std_Eff | Coef. | Std.Err. | t | P > \|t\| | [95% Conf. Interval] | |
|---------|-------|----------|---|-----------|------------------|------|
| slope | -0.112529 | 0.155612 | -0.27 | 0.478 | -0.438229 | -0.21327 |
| bise | -0.92655 | 0.74789 | -1.24 | 0.32 | -2.491901 | -0.638802 |

**NOTE:** *p*>0.05 indicating good agreement (no publication bias) and a more objective description of publication bias with numerical value.

and increases birth canal space to alleviate maternal labor pain but also helps prevent pregnancy complications. There is evidence to suggest that postpartum depression (PPD) doesn't exclusively occur after childbirth, as population-based studies indicate a similar 12% occurrence rate of depression during pregnancy. This suggests that PPD symptoms may originate during pregnancy in certain cases [43]. Furthermore, studies indicate that the antidepressant effects of exercise can persist for a period after exercise cessation [44]. Therefore, engaging in aerobic exercise during pregnancy may have a greater impact on preventing PPD compared to exercise as a treatment postpartum.

The results of the "Test for subgroup difference" indicate that there were no significant differences observed in the outcomes of the supervised vs. unsupervised subgroup (p = 0.55 > 0.05, Fig 5) and the team vs. individual subgroup (p = 0.78 > 0.05, Fig 4). Fortunately, we gain some confidence by comparing the outcomes of the "Subtotal (95%CL)". The combined effect size of team exercise [MD = -1.43; 95%CL: (-1.94 to -0.93)] is larger than individual exercise [MD = -1.28; 95%CL: (-2.23 to -0.33)] (Fig 4), and the combined effect size of the supervised exercise group [MD = -1.66; 95%CL: (-2.48 to -0.85)] is larger than the unsupervised group [MD = -1.37; 95%CL: (-1.86 to -0.88)] (Fig 5), although these differences are statistically insignificant. In fact, research has indicated that social support is a crucial factor for maintaining the mental well-being of pregnant and postpartum women, with supervised exercise and team-based exercise serving as avenues for providing effective social support [45]. For instance, team exercise could create a positive environment for maternal emotional communication, sharing of maternal emotions, enhance mothers' childbirth knowledge and skills [46]. Furthermore, the team exercise could also reduce fear of labor pains, alleviate negative emotions, improve interpersonal communication, as well as enhance self-efficacy [47]. And mothers could be quickly assisted by other peers in the event of an emergencies such as falls or other discomfort to ensure the safety of the exercise [48]. Apart from that, supervised exercise refers to physical activities guided and monitored by healthcare professionals or fitness trainers. It ensures that exercises are safe and appropriate for individual pregnant and postpartum women [49]. Engaging in supervised exercise classes or programs also provides opportunities for social interaction, which is a crucial aspect of mental well-being, and can alleviate feelings of isolation when connecting with other new mothers and professionals in a supportive environment [31,50].

This study did not yield significant differences in their outcomes, and the primary reasons for this lack of significance could be as follows: Firstly, influenced by the limitations of the meta-analysis method itself, meta-analysis derives generalized conclusions from synthesizing experimental data across all studies. Some potential differences might remain undiscovered through subgroup analysis. Therefore, based on the results of the: "Test for subgroup differences", we might not be able to accurately determine the reasons for significant differences in intervention effects in the experimental designs. Secondly, constrained by the current state of research, the studies we retrieved and ultimately included primarily focused on analyzing the efficacy of exercise intervention for postpartum depression. These analyses were carried out through randomized controlled trials comparing aerobic exercise (experimental group) with standard care (control group). The absence of randomized controlled trials directly comparing supervised exercise to unsupervised exercise, and team-based exercise to individual exercise, may also have contributed to the lack of significant differences observed in comparisons between these two subgroups. In the future, delving into these differences inherent in these exercise guidelines holds substantial importance in finding the optimal exercise intervention guidelines tailored for this specific population with the disease.

Given the inherent limitations of subgroup analysis, we have employed a network meta-analysis to investigate whether different types and volumes of exercise significantly impact the

prevention and treatment of postpartum depression. The results of the network meta-analysis emphasize that, excluding the yoga group, all other groups exhibit more favorable intervention effects relative to the control group. Pairwise comparisons among the dance, swimming, cycling/walking/running (jogging), and other exercise groups did not yield statistically significant differences, indicating that the type of aerobic exercise does not directly influence the intervention effects. Different individuals have varying preferences for types of exercise, so we speculate that experimental participants might exhibit different levels of adherence to the same exercise type. This implies that these exercise types might not directly cause differences in intervention effects; rather, patients' adherence to the prescribed exercise is the direct factor influencing intervention outcomes. Due to the lack of exercise adherence data in the existing literature, this remains a speculative viewpoint. Among the 26 studies included, only a small number mentioned exercise adherence [7,9,31–33,38,39], and even then, they primarily referred to ideal rather than actual adherence, constituting a limitation of this study.

Moderate aerobic exercise can stimulate the release of endorphins, uplift mood, and alleviate symptoms of anxiety and depression [51]. Moderate exercise also helps regulate hormone levels, improve sleep quality, enhance self-awareness and self-esteem, thereby positively impacting the alleviation of postpartum depression [52]. In the 26 studies we reviewed, the experimental exercise guidelines covered a wide range of weekly exercise frequencies, varying from 1 to 6 times, along with different exercise intensities—high, moderate, and low—and exercise durations spanning from 20 to 60 minutes. This encompassed a broad spectrum of exercise volumes. After conducting the network meta-analysis, we have obtained the following results: a comparison of SUCRA values and MD (95% CI) among the three distinct exercise frequencies demonstrated that engaging in exercise 3~4 times per week outperformed the other two frequency groups. Similarly, based on the comparative analysis of SUCRA values and MD (95% CI) among the three distinct combinations of exercise intensity and duration, it became evident that moderate intensity (35~45 minutes) yielded superior intervention effects compared to the other two combinations. Therefore, engaging in exercise 3 to 4 times per week, with a moderate exercise intensity and a duration of 35 to 45 minutes, represents a more optimal, precise, and effective planned exercise volume range.

In general, the more precise the planned exercise volume, the easier the exercise plan is to follow, making it safer for pregnant and postpartum women and theoretically leading to better intervention effects. Although there might be differences between the planned and actual exercise volumes, the planned exercise volume ensures standardized measurements within the experimental group. However, this planned exercise volume doesn't imply that everyone needs to engage in the same amount of exercise, as that is not feasible in practice. Considering the individual differences among pregnant and postpartum women, the actual exercise volume can be reasonably adjusted within the planned exercise volume range. Therefore, the aim of this network meta-analysis is to explore the impact of differences in planned exercise volume on intervention effects, in order to identify a more accurate range of planned exercise volume. This holds significant implications for shaping future exercise prescription strategies.

This study has several limitations: (1) The study primarily focuses on the inherent characteristics of exercise plans, aiming to explore the sources of heterogeneity. This inclination may result in a partial analysis of potential sources of heterogeneity. For instance, factors such as participants' age, educational level, lifestyle tendencies, exercise adherence, and exercise preferences might also influence the effects of exercise interventions on postpartum depression. (2) The categorization and discussion of the observed exercise intervention protocols in this study are based on the researchers' observation of the shared attributes within the entire exercise plans. However, constrained by the researchers' individual perceptions and experiential scope, subjectivity may have influenced the categorization outcomes to some extent.

## 5. Conclusions

Thus, taken together, the efficacy of aerobic exercise in preventing and treating postpartum depression is significant compared to standard care, with a greater emphasis on prevention. The optimal prescribed exercise volume for intervention comprises a frequency of 3~4 exercise sessions per week, moderate intensity (35~45 minutes). Currently, several uncharted factors influence the optimal intervention effect of aerobic exercise, such as the potential enhancement brought by team-based and supervised exercise. Due to the lack of significant differences in some results and the limitations of the study, the interpretation of the results still needs to be approached with caution.

Recommendations for future research directions: (1) Future studies should delineate the disparities in exercise implementation plans and exercise volume between aerobic exercise for treating postpartum depression and aerobic exercise for preventing it. (2) Design experimental studies that directly compare the effects of solitary exercise sessions versus exercise sessions with companions on postpartum depression prevention and treatment outcomes, as well as compare the impacts of supervised and unsupervised exercise processes on intervention effects.

Recommendations for the formulation and implementation of exercise guidelines: (1) In crafting exercise guidelines for pregnant and postpartum women, individualize the recommendations based on each participant's interests and physical capabilities. During the implementation phase, document participants' attendance rates, fatigue levels, and exercise completion rates. Utilize this information to judiciously adapt exercise plans. (2) During the treatment process, closely monitor participants' exercise adherence, not solely the reduction of PPD symptoms. Given that PPD patients often contend with reduced motivation for physical activity, sustaining exercise becomes challenging. Ensuring exercise adherence equates to generating actual treatment efficacy for PPD sufferers.

## Supporting information

**S1 Table. PRISMA 2020 checklist.**
(DOCX)

**S2 Table. List of raw analysis data.**
(DOCX)

**S3 Table. Complete league table.**
(DOCX)

**S1 File. Review protocol.**
(PDF)

**S2 File. Search strategy.**
(DOCX)

**S3 File. Data analysis and coding process.**
(DOCX)

**S4 File. List of sensitivity analysis data.**
(DOCX)

## Author Contributions

**Conceptualization:** Renyi Liu.

**Data curation:** Hao Xu, Xiubing Wang, Jiahui Yang.

**Formal analysis:** Hao Xu.

**Funding acquisition:** Renyi Liu.

**Investigation:** Hao Xu, Renyi Liu, Xiubing Wang, Jiahui Yang.

**Methodology:** Hao Xu, Renyi Liu.

**Project administration:** Renyi Liu.

**Resources:** Renyi Liu.

**Software:** Hao Xu.

**Supervision:** Hao Xu, Renyi Liu.

**Validation:** Hao Xu.

**Visualization:** Hao Xu.

**Writing – original draft:** Hao Xu.

**Writing – review & editing:** Hao Xu.

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
