## [Decision Letter · Decision Letter 0]

24 Mar 2023

PONE-D-23-04024Effectiveness of Aerobic Exercise in the Prevention and Treatment of Postpartum Depression: Meta-analysis and Network meta-analysisPLOS ONE

Dear Dr. Liu,

Thank you for submitting your manuscript to PLOS ONE. After careful consideration, we feel that it has merit but does not fully meet PLOS ONE’s publication criteria as it currently stands. Therefore, we invite you to submit a revised version of the manuscript that addresses the points raised during the review process.

We look forward to receiving your revised manuscript.

Kind regards,

Jayonta Bhattacharjee

Academic Editor

PLOS ONE

Journal Requirements:

   "This work was financially supported by the "Outstanding Talents Cultivation Fund" of the Central University Basic Scientific Research Fund (Grant no. CUG150607).   R.L. conceived the project and obtained funding for the field and analytical expenses. All authors (H.X., R.L., X.W., J.Y.) participated in the analysis, supervised by R.L. The manuscript was written by H.X. and R.L., with editing by R.L."

    "This work was financially supported by the "Outstanding Talents Cultivation Fund" of the Central University Basic Scientific Research Fund (Grant no.      CUG150607). "

    "This work was financially supported by the "Outstanding Talents Cultivation Fund" of the Central University Basic Scientific Research Fund (Grant no. CUG150607).   R.L. conceived the project and obtained funding for the field and analytical expenses. All authors (H.X., R.L., X.W., J.Y.) participated in the analysis, supervised by R.L. The manuscript was written by H.X. and R.L., with editing by R.L."

Reviewers' comments:

Reviewer's Responses to Questions

**Comments to the Author**

1. Is the manuscript technically sound, and do the data support the conclusions?

Reviewer #1: Partly

Reviewer #2: Yes

Reviewer #3: Yes

2. Has the statistical analysis been performed appropriately and rigorously? 

Reviewer #1: Yes

Reviewer #2: Yes

Reviewer #3: Yes

3. Have the authors made all data underlying the findings in their manuscript fully available?

Reviewer #1: Yes

Reviewer #2: Yes

Reviewer #3: Yes

4. Is the manuscript presented in an intelligible fashion and written in standard English?

Reviewer #1: Yes

Reviewer #2: Yes

Reviewer #3: Yes

5. Review Comments to the Author

Reviewer #1: I appreciate the efforts of the authors to deal with such a hot topic in today's society. However, I have some concerns:

Is the study registered? for example in prospero?

I consider that there is a lack of information in the methodological part: type of studies, language, years for the selection criteria. (much more specific)

On the other hand the results discussion and conclusions are aligned, so that is a good thing.

Reviewer #2: The research issue has some practical use, and it is examined strictly in accordance with the guidelines of the meta-analysis methodology.

1. The final section of the introduction uses literary examples to show how aerobic exercise reduces PPD depressed symptoms. There is a lot of literature, but it is not particularly connected to the study topics that were later proposed. It is advised to summarize the literature rather than simply introduce the content.

2. In the inclusion criteria, (1) the age of the subjects is unclear; (2) the postpartum depression diagnostic criteria are ambiguous; (3) it is inappropriate to include depression and depressive symptoms in the study at the same time because their outcome indicators differ. It is therefore recommended to investigate them separately；(4) There was no subject designation for depression or depressive symptoms in the Table 1.

3. What are the classification criteria for the swimming, dance, cycling/walking/running, and yoga groups, whether by intensity, skill requirement, or other classification criteria? Please elaborate in the research. Furthermore, there are many studies on yoga in the included literature, which is very different from aerobic exercise, in the included literature.

4. How should the heterogeneity of prenatal exercise be explained after subgroup analysis in the discussion?

5. Please incorporate the physiological mechanism for aerobic exercise's beneficial impact on PPD depressed symptoms.

Reviewer #3: The manuscript systematically reviewed studies of aerobic exercise for postpartum depression. The findings suggested that aerobic exercise is effective for postpartum depression. Furthermore, the results of subgroup analysis showed that the team exercise, the supervised exercise and the prenatal exercise were more beneficial in improving depressive symptoms in postpartum women. However, the authors did not mention the potential side effect/risk of aerobic exercise for this group participants.

6. PLOS authors have the option to publish the peer review history of their article (what does this mean?). If published, this will include your full peer review and any attached files.

Reviewer #1: **Yes: **Cristina Silva-Jose

Reviewer #2: **Yes: **Xing Wang

Reviewer #3: No

---

## [Author Response · Author response to Decision Letter 0]

12 Apr 2023

Dear Academic Editor and Reviewers,

Thanks for your comments of our manuscript entitled “Effectiveness of Aerobic Exercise in the Prevention and Treatment of Postpartum Depression: Meta-analysis and Network meta-analysis” (Manuscript ID PONE-D-23-04024). Those comments are all valuable and helpful for revising and improving our paper. We have discussed all comments carefully and have made conscientious revision. Below, we respond to the main comments. We also have given further detail response in the up-loaded text-revised. 

Academic Editor (Jayonta Bhattacharjee)

Comment 1. Please ensure that your manuscript meets PLOS ONE's style requirements, including those for file naming.

Response: Thank you for your comments. We have made sure that the manuscript meets PLOS ONE's style requirements, including file naming.

Comment 2: Please state what role the funders took in the study. If the funders had no role, please state: "The funders had no role in study design, data collection and analysis, decision to publish, or preparation of the manuscript." If this statement is not correct you must amend it as needed. Please include this amended Role of Funder statement in your cover letter; we will change the online submission form on your behalf.

Response: We have also included a statement about the role of the funders in the study. If the funders had no role, we will state "The funders had no role in study design, data collection and analysis, decision to publish, or preparation of the manuscript."

Comment 3: We note that you have provided funding information that is not currently declared in your Funding Statement. However, funding information should not appear in the Acknowledgments section or other areas of your manuscript. We will only publish funding information present in the Funding Statement section of the online submission form. 

Response: We apologize for providing funding information in the wrong section of the manuscript. We have ensured that funding information only appears in the Funding Statement section of the online submission form.

Comment 4: PLOS requires an ORCID ID for the corresponding author in Editorial Manager on papers submitted after December 6th, 2016. Please ensure that you have an ORCID ID and that it is validated in Editorial Manager. To do this, go to ‘Update my Information’ (in the upper left-hand corner of the main menu), and click on the Fetch/Validate link next to the ORCID field. This will take you to the ORCID site and allow you to create a new ID or authenticate a pre-existing ID in Editorial Manager.

Response: We understand that PLOS requires an ORCID ID for the corresponding author in Editorial Manager. We have made sure that the corresponding author has an ORCID ID (0000-0003~4448-5719) and that it is validated in Editorial Manager.

Comment 5: Please review your reference list to ensure that it is complete and correct. If you have cited papers that have been retracted, please include the rationale for doing so in the manuscript text, or remove these references and replace them with relevant current references. Any changes to the reference list should be mentioned in the rebuttal letter that accompanies your revised manuscript. If you need to cite a retracted article, indicate the article’s retracted status in the References list and also include a citation and full reference for the retraction notice.

Response: We have reviewed the reference list as required to ensure its completeness and accuracy, and made individual modifications in accordance with the journal's reference format requirements, such as adding the number of authors, DOIs, and web links to the references. We have not cited any retracted articles. (Page 12-16 line 398-615 of the Revised Manuscript with Track Changes).

Reviewer 1

Comment 1: Is the study registered? for example in prospero?

Response: Thank you for your positive comments. We have been registered on the official Prospero website, and the registration code is CRD42023398221 (http://www.crd.york.ac.uk/prospero/#recordDetails). The contents have been added in page 1 line 18-19 of the Revised Manuscript with Track Changes.

Comment 2: I consider that there is a lack of information in the methodological part: type of studies, language, years for the selection criteria. (much more specific)

Response: We included randomized controlled trials (RCTs) that evaluated the prevention and treatment effects of aerobic exercise on postpartum depression in women, which were published between 2000 and 2023 and published in English and Chinese languages and eligible for inclusion in the meta-analysis. The contents have been added in page 3 line 90-94 of the Revised Manuscript with Track Changes.

Reviewer 2

Comment 1: The final section of the introduction uses literary examples to show how aerobic exercise reduces PPD depressed symptoms. There is a lot of literature, but it is not particularly connected to the study topics that were later proposed. It is advised to summarize the literature rather than simply introduce the content.

Response: Thank you for your positive comments. We have formulated a comprehensive summary of the closely related references for this research topic based on the reviewer’s suggestions and have adjusted statement in page 2-3 line 66-84 of the Revised Manuscript with Track Changes.

Comment 2: In the inclusion criteria, (1) the age of the subjects is unclear; (2) the postpartum depression diagnostic criteria are ambiguous; (3) it is inappropriate to include depression and depressive symptoms in the study at the same time because their outcome indicators differ. It is therefore recommended to investigate them separately；(4) There was no subject designation for depression or depressive symptoms in the Table 1.

Response: (1) We have revised the inclusion criteria and recruited subjects who were pregnant women of an appropriate age, usually ranging from 20 to 36 years. (Page 3 line 100-102 of the Revised Manuscript with Track Changes). 

(2) Participants in the study are screened for postpartum depression (PPD) using the EPDS questionnaire, and their depressive symptoms are evaluated based on the questionnaire scores. (Page 3 line 105-106 of the Revised Manuscript with Track Changes) 

(3) We apologize for including both depression and depressive symptoms in the study, which was inappropriate. Although there is some correlation between major depression and depressive symptoms as an outcome measure, they are not identical concepts. This study is limited by the fact that both were included, which may be an important factor contributing to the high heterogeneity in the meta-analysis. Therefore, we are deeply grateful for the suggestions made by the reviewers and have made the following adjustments: After carefully reviewing the 26 studies included in our analysis, we found that none of them differentiated between participants diagnosed with postpartum depression (PPD) and those exhibiting PPD symptoms (EPDS baseline) when selecting the experimental population. We have therefore redefined our study population with a more rigorous terminology, describing them as “pregnant or postpartum women with severe or mild PPD symptoms”. In addition, we will use the EPDS scoring system as the outcome measure. (Page 3 line 100-102 of the Revised Manuscript with Track Changes)

(4) The randomized controlled trials (RCTs) included in this study recruited participants based on the severity of their postpartum depression (PPD) symptoms, as indicated by their EPDS baseline. Additional details about the EPDS baseline of the subjects included in Table 1 have been provided. (Page 8 of the Figure file; Table 1 of the Revised Figure)

Comment 3: What are the classification criteria for the swimming, dance, cycling/walking/running, and yoga groups, whether by intensity, skill requirement, or other classification criteria? Please elaborate in the research. Furthermore, there are many studies on yoga in the included literature, which is very different from aerobic exercise, in the included literature.

Response: The practice of categorizing aerobic exercises based on different skill requirements includes swimming, dancing, cycling/walking/running, and yoga. Pregnant women commonly opt for yoga as an aerobic exercise; however, network meta-analysis results demonstrate that its intervention is less effective in comparison to other aerobic exercise routines such as dance, swimming, cycling, walking, and running. It is imperative to acknowledge that the effectiveness of aerobic exercise interventions for PPD is not exclusively reliant on exercise types, amounts, or modes. As such, the intervention effect of yoga is subject to the influence of these comprehensive factors, with the precise reasons necessitating further study (Page 9-10 line 318-331 of the Revised Manuscript with Track Changes).

Comment 4: How should the heterogeneity of prenatal exercise be explained after subgroup analysis in the discussion?

Response: The heterogeneity of the prenatal exercise group has been explained and supplemented in the discussion section of the article, with the specific reasons mainly including the following two points: (1) Compared with postpartum, pregnant women had larger differences in the severity of depressive symptoms during pregnancy, resulting in larger fluctuations in EPDS scores of the included studies. (2) Through network meta-analysis, it was found that the exercise intervention schemes of the prenatal intervention group were quite different, and the exercise types, intensity-duration combinations and exercise frequencies involved would affect the effect of aerobic exercise on postpartum depressive symptoms. (Page 11 line 363-379 of the Revised Manuscript with Track Changes).

Comment 5: Please incorporate the physiological mechanism for aerobic exercise's beneficial impact on PPD depressed symptoms.

Response: We have provided a supplement in the discussion section on the physiological mechanisms of the beneficial effect of aerobic exercise on PPD depressive symptoms. (Page 10-11 line 350-362 of the Revised Manuscript with Track Changes)

Reviewer 3

Comment 1: However, the authors did not mention the potential side effect/risk of aerobic exercise for this group participants.

Response: Thank you for your positive comments. We have supplemented this in the discussion and conclusion sections: Potential adverse effects/risks that might be experienced by participants during aerobic exercise can be avoided through the control and supervision of exercise intensity. This study explores the effects of aerobic exercise interventions on PPD while ensuring that such exercise is beneficial to both mothers and fetuses, as well as minimizing potential adverse effects and risks. (Page 10 line 333-334 and Page 10-11 line 358-360 and Page 11 line 388-393 of the Revised Manuscript with Track Changes).

---

## [Decision Letter · Decision Letter 1]

28 Apr 2023

PONE-D-23-04024R1Effectiveness of Aerobic Exercise in the Prevention and Treatment of Postpartum Depression: Meta-analysis and Network meta-analysisPLOS ONE

Dear Dr. Liu,

Thank you for submitting your manuscript to PLOS ONE. After careful consideration, we feel that it has merit but does not fully meet PLOS ONE’s publication criteria as it currently stands. Therefore, we invite you to submit a revised version of the manuscript that addresses the points raised during the review process.

We look forward to receiving your revised manuscript.

Kind regards,

Jayonta Bhattacharjee

Academic Editor

PLOS ONE

Journal Requirements:

Additional Editor Comments:

I have concerns about the following minor issues. I therefore request that you revise the text to fix the grammatical errors and improve the overall readability of the text.

Authors also need to carefully read the manuscript again for the accuracy of references. Here are some minor corrections.

Line 21, Results: The line should start with word ‘Twenty-Six’ rather than 26.

Line 139-140: There is a repetition of a sentence.

Line 159-160: It seems ‘And’ in the staring of the sentence is not necessary.

Line 162: The line should start with word ‘Twenty-Six’ rather than 26.

Line 163: It should be from 2003. Reference number 23 seems from the year 2003. Please carefully check all the references.

Reviewers' comments:

Reviewer's Responses to Questions

**Comments to the Author**

1. If the authors have adequately addressed your comments raised in a previous round of review and you feel that this manuscript is now acceptable for publication, you may indicate that here to bypass the “Comments to the Author” section, enter your conflict of interest statement in the “Confidential to Editor” section, and submit your "Accept" recommendation.

Reviewer #2: (No Response)

Reviewer #3: All comments have been addressed

2. Is the manuscript technically sound, and do the data support the conclusions?

Reviewer #2: Yes

Reviewer #3: Yes

3. Has the statistical analysis been performed appropriately and rigorously? 

Reviewer #2: Yes

Reviewer #3: Yes

4. Have the authors made all data underlying the findings in their manuscript fully available?

Reviewer #2: Yes

Reviewer #3: Yes

5. Is the manuscript presented in an intelligible fashion and written in standard English?

Reviewer #2: Yes

Reviewer #3: Yes

6. Review Comments to the Author

Reviewer #2: (No Response)

Reviewer #3: The authors addressed my comment well. The manuscript is well written with clear methodology and reasonable analysis, the results and interpretation are also well organized. I have no further comment.

7. PLOS authors have the option to publish the peer review history of their article (what does this mean?). If published, this will include your full peer review and any attached files.

Reviewer #2: No

Reviewer #3: **Yes: **LIN Jingxia Jessie

---

## [Author Response · Author response to Decision Letter 1]

30 Apr 2023

Thanks for your comments of our manuscript entitled “Effectiveness of Aerobic Exercise in the Prevention and Treatment of Postpartum Depression: Meta-analysis and Network meta-analysis” (Manuscript ID PONE-D-23-04024R1). Those comments are all valuable and helpful for revising and improving our paper. We have discussed all comments carefully and have made conscientious revision. Below, we respond to the main comments. We also have given further detail response in the up-loaded text-revised. 

Academic Editor (Jayonta Bhattacharjee)

Comment 1. Please review your reference list to ensure that it is complete and correct.

Response: Thank you for your comments. We have reviewed the reference list as required to ensure its completeness and accuracy, and made individual modifications in accordance with the journal’s reference format requirements, such as adding the number of authors, DOIs, and web links to the references. We have not cited any retracted articles.

Comment 2: I have concerns about the following minor issues. I therefore request that you revise the text to fix the grammatical errors and improve the overall readability of the text. Authors also need to carefully read the manuscript again for the accuracy of references. Here are some minor corrections.

Line 21, Results: The line should start with word ‘Twenty-Six’ rather than 26.

Line 139-140: There is a repetition of a sentence.

Line 159-160: It seems ‘And’ in the staring of the sentence is not necessary.

Line 162: The line should start with word ‘Twenty-Six’ rather than 26.

Line 163: It should be from 2003. Reference number 23 seems from the year 2003. Please carefully check all the references.

Response: We deeply apologize for the lack of rigor in the editing of the submitted paper and have made careful revisions based on all the suggestions made by the academic editor (Jayonta Bhattacharjee). (Line 21, Line 134-135, Line 152-153, Line 156 and Line 158 of the Revised Manuscript with Track Changes).

---

## [Decision Letter · Decision Letter 2]

18 May 2023

PONE-D-23-04024R2Effectiveness of Aerobic Exercise in the Prevention and Treatment of Postpartum Depression: Meta-analysis and Network meta-analysisPLOS ONE

Dear Dr. Liu,

Thank you for submitting your manuscript to PLOS ONE. After careful consideration, we feel that it has merit but does not fully meet PLOS ONE’s publication criteria as it currently stands. Therefore, we invite you to submit a revised version of the manuscript that addresses the points raised during the review process.Please revise the manuscript according to the reviewer comments. 

We look forward to receiving your revised manuscript.

Kind regards,

Jayonta Bhattacharjee

Academic Editor

PLOS ONE

Journal Requirements:

Reviewers' comments:

Reviewer's Responses to Questions

**Comments to the Author**

1. If the authors have adequately addressed your comments raised in a previous round of review and you feel that this manuscript is now acceptable for publication, you may indicate that here to bypass the “Comments to the Author” section, enter your conflict of interest statement in the “Confidential to Editor” section, and submit your "Accept" recommendation.

Reviewer #4: (No Response)

2. Is the manuscript technically sound, and do the data support the conclusions?

Reviewer #4: Yes

3. Has the statistical analysis been performed appropriately and rigorously? 

Reviewer #4: Yes

4. Have the authors made all data underlying the findings in their manuscript fully available?

Reviewer #4: Yes

5. Is the manuscript presented in an intelligible fashion and written in standard English?

Reviewer #4: Yes

6. Review Comments to the Author

Reviewer #4: Although some subgroup interventions did not yield significant comparisons, taken together, the results demonstrated that the aerobic exercise intervention is an effective tool for the prevention and treatment of PPD. Also, the effect of aerobic exercise on PPD stems from the combined effect of multiple variables within the exercise prescription. The investigators also examined multiple possibilities which result in the SUCRA scores and plots for a network meta-analysis examination of the results.

The paper is well presented and the results appear to follow from all the elements performed given the systematic review and meta-analysis. Most of the edits required by the authors have been incorporated into the manuscript. However, there do remain some minor edits to be addressed. For example, on page 5, line 168 the word, ‘ridk’ should be ‘risk’.

7. PLOS authors have the option to publish the peer review history of their article (what does this mean?). If published, this will include your full peer review and any attached files.

Reviewer #4: No

---

## [Author Response · Author response to Decision Letter 2]

30 May 2023

Thanks for your comments of our manuscript entitled “Effectiveness of Aerobic Exercise in the Prevention and Treatment of Postpartum Depression: Meta-analysis and Network meta-analysis” (Manuscript ID PONE-D-23-04024R2). Those comments are all valuable and helpful for revising and improving our paper. We have discussed all comments carefully and have made conscientious revision. Below, we respond to the main comments. We also have given further detail response in the up-loaded text-revised. 

Academic Editor (Jayonta Bhattacharjee)

Comment 1. Please review your reference list to ensure that it is complete and correct.

Response: Thank you for your comments. We have reviewed the reference list as required to ensure its completeness and accuracy, and made individual modifications in accordance with the journal’s reference format requirements, such as adding the number of authors, DOIs, and web links to the references (Page 13 line 473-476 and Page 14 line 511-516 and Page 15 line 561-564 of the Revised Manuscript with Track Changes). We have not cited any retracted articles. 

Reviewer 4

Comment 1: Although some subgroup interventions did not yield significant comparisons, taken together, the results demonstrated that the aerobic exercise intervention is an effective tool for the prevention and treatment of PPD. Also, the effect of aerobic exercise on PPD stems from the combined effect of multiple variables within the exercise prescription. The investigators also examined multiple possibilities which result in the SUCRA scores and plots for a network meta-analysis examination of the results. The paper is well presented and the results appear to follow from all the elements performed given the systematic review and meta-analysis. Most of the edits required by the authors have been incorporated into the manuscript. However, there do remain some minor edits to be addressed. For example, on page 5, line 168 the word, ‘ridk’ should be ‘risk’.

Response: We sincerely appreciate your review and valuable feedback on our manuscript. We have carefully considered your comments and have incorporated most of the edits into the revised version of the paper. We deeply apologize for the oversight regarding the word ‘rick’ on page 5, line 168. We have made the necessary correction, replacing it with ‘risk’ as suggested. We also modified some punctuation mark in the article.

We are delighted to hear that you found our study’s overall results to demonstrate the efficacy of aerobic exercise intervention as an effective tool for the prevention and treatment of postpartum depression (PPD). We acknowledge that while some subgroup interventions did not yield significant comparisons, when considered collectively, the results provide strong evidence supporting the effectiveness of aerobic exercise intervention for PPD. Furthermore, we agree with your assessment that the effect of aerobic exercise on PPD is influenced by multiple variables within the exercise prescription. We also appreciate your recognition of the comprehensive approach we employed, examining various possibilities, which led to the SUCRA scores and plots utilized in the network meta-analysis of the results. 

Once again, we extend our heartfelt thanks for your time and valuable input.

---

## [Decision Letter · Decision Letter 3]

8 Aug 2023

PONE-D-23-04024R3Effectiveness of Aerobic Exercise in the Prevention and Treatment of Postpartum Depression: Meta-analysis and Network meta-analysisPLOS ONE

Dear Dr. Liu,

Thank you for submitting your manuscript to PLOS ONE. After careful consideration, we feel that it has merit but does not fully meet PLOS ONE’s publication criteria as it currently stands. Therefore, we invite you to submit a revised version of the manuscript that addresses the points raised during the review process.

We look forward to receiving your revised manuscript.

Kind regards,

Jayonta Bhattacharjee

Academic Editor

PLOS ONE

A**dditional Editor Comments:**

As you are aware, concerns have been raised regarding the contents of the submission after the accept decision was issued. The submission has been re-evaluated as a result of the concerns raised, and the comments provided by the reviewer can be found below. Please note that some of these concerns have been discussed with you previously, and some may be new. At this time, we request that you comprehensively revise the submission to address all concerns raised.

Please note that the revised manuscript will be reviewed, and we cannot guarantee any specific editorial outcome.

Reviewers' comments:

Reviewer's Responses to Questions

**Comments to the Author**

1. If the authors have adequately addressed your comments raised in a previous round of review and you feel that this manuscript is now acceptable for publication, you may indicate that here to bypass the “Comments to the Author” section, enter your conflict of interest statement in the “Confidential to Editor” section, and submit your "Accept" recommendation.

Reviewer #4: All comments have been addressed

Reviewer #5: (No Response)

2. Is the manuscript technically sound, and do the data support the conclusions?

Reviewer #4: (No Response)

Reviewer #5: No

3. Has the statistical analysis been performed appropriately and rigorously? 

Reviewer #4: (No Response)

Reviewer #5: I Don't Know

4. Have the authors made all data underlying the findings in their manuscript fully available?

Reviewer #4: (No Response)

Reviewer #5: No

5. Is the manuscript presented in an intelligible fashion and written in standard English?

Reviewer #4: (No Response)

Reviewer #5: Yes

6. Review Comments to the Author

Reviewer #4: (No Response)

Reviewer #5: This manuscript is an interesting and novel meta-analysis of randomized controlled trials investigating aerobic exercise for people with postpartum depression. The topic is important as many people experience postpartum depression, which causes significant negative effects to the person themselves and their relationship with their child. Exercise represents an easily accessible and safe alternative to antidepressant medication. Though this manuscript is strong in overall significance and novelty, there are several major concerns about the methodology and interpretation of results that limit the contribution to the literature. Of most importance is the interpretation of the subgroup analyses, many of which do not show significant differences between groups but appear to be interpreted as such in the abstract, results, and discussion. These interpretations are misleading regarding conclusions to be drawn from the data. Please see below for specific major and minor comments by section. As noted below, there are several incorrect results when comparing the text vs. the tables/figures. Please review for accuracy.

Abstract

1. The background section seems to indicate this meta-analysis is focusing on exercise during pregnancy but some studies included also examine exercise postpartum so this should be clarified.

2. In Figure 3, the overall effect appears to be -1.90 but is reported as -2.36 (similar error in results section).

3. The first sentence of the results should indicate the comparison group (i.e., postpartum people in non-exercise interventions or treatment as usual).

4. The report of subgroup analysis results is misleading. Overall effects of each subgroup are reported instead of the comparison between subgroups. Therefore, the claim that these subgroups are “more beneficial” cannot be made as the subgroups are not significantly different from each other. If the overall effect of one subgroup is reported, the other subgroup should also be included. For example, overall effects of the individual exercise subgroup were also significant as compared to control, so there is little justification for promoting only the value of team based exercise. It would be best practice to report the test for subgroup differences instead of overall effects of the individual subgroups.

5. If reporting the overall effect for supervised exercise, it seems the p value is incorrect (from the heterogeneity test).

6. The final sentence of the results should note that this is in comparison to other frequencies/intensities.

7. As noted for the results section, the conclusion regarding subgroups is misleading given lack of significant difference.

8. The final sentence suggests that the benefits are just during pregnancy—is this the case for all the studies included in the comparisons? Or were some postpartum?

Introduction

1. Line 46: The reported prevalence range is very large, which decreases the meaningfulness of the statement. Is there a prevalence that may be most accurate? Or it would be helpful to explain this wide range.

2. The final two sentences of the introduction would be better suited for the discussion. Instead, it would be useful to include a priori hypotheses.

Methods

1. What is the rationale for the timeframe for the publications being after 2000?

2. What is the rationale for the need to include both “postpartum depression” AND “maternal depression” in title or abstract. I would imagine this might exclude some studies that had one or the other terms.

3. What is the rationale for the age range up to 36 years old?

4. Were moderate depression symptoms included? Only mild or severe are indicated in inclusion criteria.

5. The EPDS is referred to differently in lines 101 (P=postpartum) vs. 120/122 (postnatal).

6. Statistical analysis/figures: It is unclear to me what the mean/SD for the experimental group represents vs. mean/SD for control group. Are these mean scores at post-exercise intervention? Mean changes from pre to post intervention? In the statistical analysis section, it reports “comparing scores before and after the aerobic exercise intervention.” It should be clarified if the means being compared are mean changes from pre to post or something else.

Results

1. As noted previously, the overall effect (line 171) appears to be misreported if the numbers in Figure 3 are correct.

2. Section 3.3.1 should note the comparison group for the effect

3. Please adjust to be a complete sentence: The combined effect size for the aerobic exercise effect on postpartum depression intervention was (26 RCTs; MD= -2.36, 95% CL: -2.58 to -2.15; p<0.00001).

4. For all subgroup analyses and network analyses, please indicate what “other intervening variables” mean.

5. As noted previously, subgroup results are misleading as reported. The tests for subgroup differences indicate no significant differences in team vs. individual exercise (p=.78) or supervised vs. unsupervised exercise (p=.55). There is, however, a significant difference in prenatal vs. postnatal exercise (p=.02). To be most transparent, these subgroup differences should be reported in the results if authors want to say “more beneficial” or “less beneficial” as those are the appropriate tests for those conclusions. Authors may also choose to report the overall effect for each subgroup and relative direction of the effect but should not make the claim that one subgroup is “more beneficial” if the subgroup analysis is not significant. This type of conclusion is especially misleading for team vs. individual exercise in which both subgroups have significant overall effects as compared to controls, which ultimately means either could be recommended as a good source of exercise to reduce postpartum symptoms. This can be similarly said of supervised vs. unsupervised exercise as well.

6. Incorrect p values are reported for supervised and unsupervised exercise overall effects.

7. It is unclear why cycling, walking, and running are grouped together. Walking and running/cycling would likely have different intensities and ultimately different effects (especially since intensity did yield differential results).

8. For exercise frequency, was this defined as the actual frequency with which individuals completed exercise or the prescribed frequency? If the latter, though this frequency may be prescribed, it is likely that many individuals within the study did not meet this frequency. This should be clarified and noted within the discussion/limitations if the latter. Similar note for exercise intensity in terms of actual vs. prescribed.

9. For ease of understanding, exercise intensities should also be described in terms of low, moderate, high.

10. How were the exercise intensity/duration groupings made?

11. In section 3.6, p for publication bias is listed as .23 but Table 6 says .32.

Discussion

1. As noted previously, given lack of significant differences between subgroups, team exercise and supervised exercise cannot be described as increasing effectiveness of the interventions. If the discussion is to theorize why team exercise had a better overall effect than control, this is fine, but should be clearly stated. Additionally, since individual exercise also had better effect than control, it should equally be included in the discussion for possible mechanisms. This is similar re: supervised vs. unsupervised exercise given significant overall effects of the individual subgroups and lack of significant differences between the subgroups.

2. In general, the discussion is far too prescriptive.

3. Physiological mechanisms are proposed and include psychological components. If including psychological components (e.g., bullet 3), the vast literature on mood benefits should also be noted.

4. Limitations are minimally mentioned and the discussion of limitations should be expanded.

Conclusions

1. Similar notes to previously stated re: interpretations of subgroup analyses and being far too prescriptive (i.e., not balancing potential limitations of the data).

2. Expansion of future directions would be beneficial in discussion (and possibly conclusions).

Tables

1. Can Table 1 be put in landscape? It is very hard to follow as written.

2. For the EPDS, what does the mean represent? Is this the mean at the end of the intervention? Or mean change? Mean change would be more ideal as this would account for baseline levels of depression.

3. What is the rationale for categorizing exercise classes as “team”? Is this better defined as individual vs. group? It may be helpful to define these terms in the methods.

4. The inclusion criteria note that age range should be 20-36; however, it appears mean age for Yan is 36.6?

7. PLOS authors have the option to publish the peer review history of their article (what does this mean?). If published, this will include your full peer review and any attached files.

Reviewer #4: No

Reviewer #5: No

---

## [Author Response · Author response to Decision Letter 3]

29 Aug 2023

Dear Reviewer,

We would like to express our gratitude for your valuable comments on our manuscript titled "Effectiveness of Aerobic Exercise in the Prevention and Treatment of Postpartum Depression: Meta-analysis and Network meta-analysis" (Revision required [PONE-D-23-04024R3]- [EMID: a8940538520f27ef]). These comments have greatly contributed to the revision and improvement of our manuscript. We have diligently considered all the comments and made substantial revisions accordingly. In the following sections, we address the major comments. Additionally, we have provided more detailed responses in the uploaded revised version of the manuscript.

Abstract

1. The background section seems to indicate this meta-analysis is focusing on exercise during pregnancy but some studies included also examine exercise postpartum so this should be clarified.

Response: In light of this matter, we have made revisions to the background section. The exercise interventions employed in this study serve two purposes: one is to prevent postpartum depression, while the other focuses on treating postpartum depression, without specifying a specific exercise period. It's worth noting that this study encompasses both pregnant women and postpartum women as subjects. As a result, the timing of exercise for this population could occur during pregnancy, postpartum, or even both. Consequently, specifying a particular exercise period for pregnant and postpartum women could lead to confusion. Taking into account the aforementioned issues, we have made appropriate adjustments. Therefore, the expression here was indeed incorrect. It has now been revised to align with the overall context of the manuscript. (Page 1, line 9-17).

2. In Figure 3, the overall effect appears to be -1.90 but is reported as -2.36 (similar error in results section).

Response: The data has been revised and is consistent with the charts and figures. (Page 1, line 33)

3. The first sentence of the results should indicate the comparison group (i.e., postpartum people in non-exercise interventions or treatment as usual).

Response: We have improved the sentences in accordance with the above requests. (Page 1, line 30-31)

4. The report of subgroup analysis results is misleading. Overall effects of each subgroup are reported instead of the comparison between subgroups. Therefore, the claim that these subgroups are “more beneficial” cannot be made as the subgroups are not significantly different from each other. If the overall effect of one subgroup is reported, the other subgroup should also be included. For example, overall effects of the individual exercise subgroup were also significant as compared to control, so there is little justification for promoting only the value of team based exercise. It would be best practice to report the test for subgroup differences instead of overall effects of the individual subgroups.

Response: The manuscript has been revised to incorporate the significance results of "Test for subgroup difference" and to restate the results of subgroup analysis by considering the combined effect sizes (Subtotal 95%CL) of each subgroup. For example: “Subgroup analysis suggests that the intervention objective (prevention vs. treatment) of exercise could potentially be a source of heterogeneity in this study，as the “Test for subgroup difference” revealed the presence of significant distinctions (p=0.02＜0.05). The “Test for subgroup difference” yielded non-significant results for both the supervised vs. unsupervised subgroup comparison (p=0.55 > 0.05) and the individual vs. team subgroup comparison (p=0.78 > 0.05). Nonetheless, when assessing their effect sizes [Subtotal (95%CL)], the supervised exercise group [-1.66 (-2.48, -0.85)] exhibited a slightly better performance than the unsupervised exercise group [-1.37 (-1.86, -0.88)], while the team exercise group [-1.43 (-1.94, -0.93)] slightly outperformed the individual exercise group [-1.28 (-2.23, -0.33)].” (Page 2, line 34-42)

5. If reporting the overall effect for supervised exercise, it seems the p value is incorrect (from the heterogeneity test).

Response: The manuscript has been uniformly revised in terms of expressing heterogeneity and effect sizes. In Figure 5, a total of 4 p-values are presented, representing the heterogeneity within the supervised exercise group, the heterogeneity within the unsupervised exercise group, the overall heterogeneity across the two groups of studies, and finally, the statistical significance of the "Test for subgroup difference" regarding the differences between the two groups' outcomes. In the heterogeneity assessment of the Meta-analysis, the p-value and the meaning represented by I2 for the "supervised exercise" group are consistent. They both indicate whether there is statistical significance in the variability of study results within a specific subgroup, rather than indicating the significance testing between the experimental and control groups in all studies. The summarized results of the experimental and control groups for all studies are represented by Overall Effects (MD) and 95%CL, presented intuitively through a forest plot. For this subgroup, the p-value is 0.12 > 0.05 from the heterogeneity test of subgroup, and I2 is 37% < 50%；the overall effect for supervised exercise is presented as (Subtotal 95%CL), and corresponding p-values are not provided. Instead, the significance of intervention effects is determined by whether the 95% confidence interval (95%CL) includes 0. This indicates that the variability of results within all studies included in this subgroup is not statistically significant, and it can also be considered to have good consistency. To avoid misinterpretation, the heterogeneity across all studies is indicated using the I2 statistic，subgroup effect sizes are all presented as "MD 95%CL."(Page 2, line 34-40)

6. The final sentence of the results should note that this is in comparison to other frequencies/intensities.

Response: The requested changes have been made. (Page 2, line 46-53)

7. As noted for the results section, the conclusion regarding subgroups is misleading given lack of significant difference.

Response: The subgroup analysis results have been reinterpreted by incorporating the significance results of the "Test for subgroup difference" and the combined effect sizes (Subtotal 95%CL) of each subgroup. (Page 2, line 64-66)

8. The final sentence suggests that the benefits are just during pregnancy—is this the case for all the studies included in the comparisons? Or were some postpartum?

Response: In fact, this study focuses on both pregnant women and postpartum women as subjects. Therefore, the timing of exercise for pregnant and postpartum women could be during pregnancy, postpartum, or even both. Consequently, we should not specify the timing of exercise for this population to avoid confusion. We have made new adjustments considering the issues mentioned earlier. (Page 2, line 61-62)

Introduction

1. Line 46: The reported prevalence range is very large, which decreases the meaningfulness of the statement. Is there a prevalence that may be most accurate? Or it would be helpful to explain this wide range.

Response: The optimal revisions have been made according to the reviewer's suggestions. (Page 3, line 77-80)

2. The final two sentences of the introduction would be better suited for the discussion. Instead, it would be useful to include a priori hypotheses.

Response: The most appropriate modifications have been made as per the reviewer's suggestions. (Page 4, line 110-117)

Methods

1. What is the rationale for the timeframe for the publications being after 2000?

Response: We realized that setting the search start date to the year 2000 could potentially lead to the omission of some studies. Therefore, we conducted a search and update of literature from the inception of the databases, but did not find any studies that met the inclusion criteria before the year 2000. As a result, we are now changing the literature search time frame to the inception of the databases. (Page 4, line 126-130)

2. What is the rationale for the need to include both “postpartum depression” AND “maternal depression” in title or abstract. I would imagine this might exclude some studies that had one or the other terms.

Response: After examining the detailed search strategy in the supplementary materials, we have identified writing oversights in the manuscript regarding the information in this section. Here, in fact, it should be "OR." The error has already been corrected in the manuscript. (Page 4, line 131)

3. What is the rationale for the age range up to 36 years old?

Response: Originally, the ideal age range for the inclusion criteria of the study participants was intended to be within the age range of eligible pregnant and postpartum women. However, due to the limited number of literatures meeting these criteria, the age range was eventually narrowed down to participants aged 18 and above. Unfortunately, during the writing process, the study editor did not promptly update the inclusion criteria, resulting in an error due to outdated information. The inclusion criteria for participant age have now been corrected to be 18 years and above (adult females). (Page 4, line 137)

4. Were moderate depression symptoms included? Only mild or severe are indicated in inclusion criteria.

Response: Due to the same reason as in Question 3, the intended meaning of this English sentence does not accurately convey the author's final intention. The entire sentence has been revised to: “The participants are normal pregnant women or postpartum depression patients who are adults (≥18 years).” (Page 4, line 136-137)

5. The EPDS is referred to differently in lines 101 (P=postpartum) vs. 120/122 (postnatal).

Response: The use of this term has been standardized throughout “postnatal”. (Page 5, line 142)

6. Statistical analysis/figures: It is unclear to me what the mean/SD for the experimental group represents vs. mean/SD for control group. Are these mean scores at post-exercise intervention? Mean changes from pre to post intervention? In the statistical analysis section, it reports “comparing scores before and after the aerobic exercise intervention.” It should be clarified if the means being compared are mean changes from pre to post or something else.

Response: In reality, the term "outcome data" here refers to the scores on the Edinburgh Postnatal Depression Scale for the experimental and control groups after exercise interventions, represented by means and standard deviations. Based on this data, we conducted Meta-analysis to calculate the effect sizes (MD) and 95% confidence intervals for the experimental and control groups in each study. By aggregating all these results, we can determine whether exercise intervention is effective in preventing and treating postpartum depression. This determination relies on whether there is a significant difference in the outcomes between the experimental and control groups in the Meta-analysis. The erroneous sentence concerning the expression of outcome data for the experimental and control groups after exercise intervention has been corrected. To enhance understanding, the explanation and clarification of the data analysis have been reinforced. (Page 6, line 210-217)

Results

1. As noted previously, the overall effect (line 171) appears to be misreported if the numbers in Figure 3 are correct.

Response: The numerical values in the text have been corrected according to the data in the figures. (Page 9-10, line 260-263)

2. Section 3.3.1 should note the comparison group for the effect

Response: The requested supplementation has been completed. (Page 9-10, line 263-263)

3. Please adjust to be a complete sentence: The combined effect size for the aerobic exercise effect on postpartum depression intervention was (26 RCTs; MD= -2.36, 95% CL: -2.58 to -2.15; p<0.00001).

Response: The requested improvements have been made as per the requirements. (Page 9-10, line 260-261)

4. For all subgroup analyses and network analyses, please indicate what “other intervening variables” mean.

Response: “other intervening variables” This indicates that factors other than the grouping variable need to be controlled to account for potential influences on the outcomes. This can be achieved through methods such as random allocation and adjustments through matching. For instance, in the context of the grouping variable comparing supervised vs. unsupervised, it is important to rigorously control other potential influencing factors between the two groups, such as the intervention purpose (prevention vs. treatment), exercise volume (type, intensity, duration, frequency), and exercise organization (team vs. individual). Random allocation ensures that factors other than supervision itself are balanced between the supervised and unsupervised groups, with strict differentiation solely based on the "supervision" factor. This portion of content has been added to the Grouping Methods section as requested. (Page 10, line 272-273,286-287,298-299,312-313,336-337,355-356,)

5. As noted previously, subgroup results are misleading as reported. The tests for subgroup differences indicate no significant differences in team vs. individual exercise (p=.78) or supervised vs. unsupervised exercise (p=.55). There is, however, a significant difference in prenatal vs. postnatal exercise (p=.02). To be most transparent, these subgroup differences should be reported in the results if authors want to say “more beneficial” or “less beneficial” as those are the appropriate tests for those conclusions. Authors may also choose to report the overall effect for each subgroup and relative direction of the effect but should not make the claim that one subgroup is “more beneficial” if the subgroup analysis is not significant. This type of conclusion is especially misleading for team vs. individual exercise in which both subgroups have significant overall effects as compared to controls, which ultimately means either could be recommended as a good source of exercise to reduce postpartum symptoms. This can be similarly said of supervised vs. unsupervised exercise as well.

Response: The subgroup analysis section has been accurately summarized based on the comprehensive results of the "Test for subgroup difference" and "Subtotal 95%CL." For example, Subgroup analysis suggests that the intervention objective (prevention vs. treatment) of exercise could potentially be a source of heterogeneity in this study，as the “Test for subgroup difference” revealed the presence of significant distinctions (p=0.02＜0.05). The “Test for subgroup difference” yielded non-significant results for both the supervised vs. unsupervised subgroup comparison (p=0.55 > 0.05) and the individual vs. team subgroup comparison (p=0.78 > 0.05). Nonetheless, when assessing their effect sizes [Subtotal (95%CL)], the supervised exercise group [-1.66 (-2.48, -0.85)] exhibited a slightly better performance than the unsupervised exercise group [-1.37 (-1.86, -0.88)], while the team exercise group [-1.43 (-1.94, -0.93)] slightly outperformed the individual exercise group [-1.28 (-2.23, -0.33)]. (Page 10, line 275-277,287-293,201-306,)

6. Incorrect p values are reported for supervised and unsupervised exercise overall effects.

Response: Here, we should focus on reporting whether the differences between subgroups are statistically significant, specifically the p-values of “Test for subgroup difference”. The error in reporting at this point has been corrected. The overall effect for supervised exercise is presented as (Subtotal 95%CL), and corresponding p-values are not provided. Instead, the significance of intervention effects is determined by whether the 95% confidence interval (95%CL) includes 0. (Page 10, line 287-293)

7. It is unclear why cycling, walking, and running are grouped together. Walking and running/cycling would likely have different intensities and ultimately different effects (especially since intensity did yield differential results).

Response: The methodology section (Added Section 2.4. Grouping criteria) of the manuscript has been explained：“An elucidation is warranted for categorizing cycling/walking/running as a singular exercise type. This decision is attributed to the relatively infrequent occurrence of these activities in isolation. Exercise guidelines often combine cycling/walking or running (jogging) /walking within training plans. Additionally, by considering the weekly exercise volume involving all these aerobic exercises, the groups with similar weekly exercise volume are classified under yoga, dance, and swimming categories, while the less frequent exercise types are grouped together.” (Page 6, line 191-197)

8. For exercise frequency, was this defined as the actual frequency with which individuals completed exercise or the prescribed frequency? If the latter, though this frequency may be prescribed, it is likely that many individuals within the study did not meet this frequency. This should be clarified and noted within the discussion/limitations if the latter. Similar note for exercise intensity in terms of actual vs. prescribed.

Response: The discussed exercise volume (exercise frequency, type, duration, intensity) in this study refers to the prescribed exercise volume in the exercise program, rather than the actual completed exercise volume by the participants. This clarification has been added to the discussion section of the manuscript. This study lacks relevant data on exercise adherence among the experimental participants, thus preventing us from determining the actual completion status of the intervention protocols. Our current discussion solely focuses on the prescribed exercise volume within the exercise plans, rather than the actual exercise volume accomplished by the participants. Whether it’s the prescribed exercise volume or the actual achieved volume, both ultimately need to be tailored according to the physical capabilities and conditions of the pregnant and postpartum women. The planned exercise volume serves as a reference for safe exercise, and holds significance and value. Therefore, engaging in exercise 3 to 4 times per week, along with moderate exercise intensity and a duration of 35 to 45 minutes, represents an optimal prescribed exercise volume, offering guidance for exercise prescription. However, it may not necessarily reflect the actual exercise volume achieved by all participants. (Page 16, line 479-489)

9. For ease of understanding, exercise intensities should also be described in terms of low, moderate, high.

Response: The wording related to exercise intensity has been changed as requested: “low exercise intensity: 40% HRR; moderate exercise intensity: 50%~ 60% HRR; high exercise intensity: 65% ~ 74% HRR” (Page 13, line 372)

10. How were the exercise intensity/duration groupings made?

Response: The methodology section (Added Section 2.4. Grouping criteria) has been supplemented with the relevant explanations: “Exercise volume are arranged in three levels, descending from high to low volume. Based on exercise frequency, they are stratified into three categories: 2~3 times per week, 3~4 times per week, and 5-6 times per week. Notably, upon scrutinizing the planned exercise intensities and durations across all the included studies, a pattern emerges where higher exercise intensities are often paired with shorter exercise durations. Aligning similar patterns of exercise intensity and duration culminates in three categories: high intensity (50~60 minutes), moderate intensity (35~45 minutes), and low intensity (20~30 minutes).” (Page 6, line 202-206)

11. In section 3.6, p for publication bias is listed as .23 but Table 6 says .32.

Response: The numerical values have been updated and are consistent with Table 5 (Page 13, line 386)

.

Discussion

1. As noted previously, given lack of significant differences between subgroups, team exercise and supervised exercise cannot be described as increasing effectiveness of the interventions. If the discussion is to theorize why team exercise had a better overall effect than control, this is fine, but should be clearly stated. Additionally, since individual exercise also had better effect than control, it should equally be included in the discussion for possible mechanisms. This is similar re: supervised vs. unsupervised exercise given significant overall effects of the individual subgroups and lack of significant differences between the subgroups.

Response: The results of the subgroup analysis have been comprehensively supplemented and improved. Consequently, the discussion section has been completely rewritten to reflect the new information. The main points covered include the reasons for the lack of significant differences in subgroups, the implications of the network meta-analysis regarding prescribed exercise volume, and the limitations of this study. (Page 14, line 399-410)

2. In general, the discussion is far too prescriptive.

Response: Regarding the main concerns of this study, the discussion section has been reanalyzed comprehensively and in-depth. (Page 14-16, line 391-498)

3. Physiological mechanisms are proposed and include psychological components. If including psychological components (e.g., bullet 3), the vast literature on mood benefits should also be noted.

Response: Due to the realization that the current focus of the discussion is not suitable for including discussions about the physiological mechanisms, the section related to the mechanisms of exercise intervention for postpartum depression, along with relevant references, has been ultimately removed. If the reviewing experts find this part necessary, we can consider adding it back based on their suggestions. However, we have noted the vast literature on mood benefits and have also added relevant references. (Page 15, line 420-433)

4. Limitations are minimally mentioned and the discussion of limitations should be expanded.

Response: The limitations of this study have been analyzed throughout the entire discussion section. (Page 15-16, line 449-463,490-498)

Conclusions

1. Similar notes to previously stated re: interpretations of subgroup analyses and being far too prescriptive (i.e., not balancing potential limitations of the data).

Response: Regarding this issue, the discussion section has been comprehensively and thoroughly analyzed. (Page 19, line 591-592)

2. Expansion of future directions would be beneficial in discussion (and possibly conclusions).

Response: Based on the suggestions provided, the discussion section has undergone a comprehensive and in-depth analysis. (Page 19, line 593-607)

Tables

1. Can Table 1 be put in landscape? It is very hard to follow as written.

Response: The positioning of figures and tables can be adjusted as per the preferences of the journal's production department. 

2. For the EPDS, what does the mean represent? Is this the mean at the end of the intervention? Or mean change? Mean change would be more ideal as this would account for baseline levels of depression.

Response: This is the mean at the end of the intervention. The term "outcome data" here refers to the scores on the Edinburgh Postnatal Depression Scale for the experimental and control groups after exercise interventions, represented by means and standard deviations. We conducted Meta-analysis based on this data, which allowed us to calculate the effect sizes and 95% confidence intervals for the experimental and control groups in each study. These calculations help determine the intervention effect resulting from exercise intervention, comparing the experimental and control groups. (Page 9, line 249-251)

3. What is the rationale for categorizing exercise classes as “team”? Is this better defined as individual vs. group? It may be helpful to define these terms in the methods.

Response: All grouping criteria have been described in the methodology section (Added Section 2.4. Grouping criteria) of this study. (Page 5, line 178-183)

4. The inclusion criteria note that age range should be 20-36; however, it appears mean age for Yan is 36.6?

Response: Regarding this issue, adjustments have already been made to the inclusion criteria. (Page 5, line 168)

We firmly believe that these revisions further enhance the quality and readability of the manuscript. Once again, we extend our heartfelt thanks for your time and valuable input. Should you have any further questions or requests, please do not hesitate to contact us. We also modified some punctuation mark in the manuscript.

Best regards,

Hao Xu, Renyi Liu

---

## [Decision Letter · Decision Letter 4]

22 Sep 2023

PONE-D-23-04024R4Effectiveness of Aerobic Exercise in the Prevention and Treatment of Postpartum Depression: Meta-analysis and Network meta-analysisPLOS ONE

Dear Dr. Liu,

Thank you for submitting your manuscript to PLOS ONE. After careful consideration, we feel that it has merit but does not fully meet PLOS ONE’s publication criteria as it currently stands. Therefore, we invite you to submit a revised version of the manuscript that addresses the points raised during the review process.

We look forward to receiving your revised manuscript.

Kind regards,

Jayonta Bhattacharjee

Academic Editor

PLOS ONE

Journal Requirements:

Reviewers' comments:

Reviewer's Responses to Questions

**Comments to the Author**

1. If the authors have adequately addressed your comments raised in a previous round of review and you feel that this manuscript is now acceptable for publication, you may indicate that here to bypass the “Comments to the Author” section, enter your conflict of interest statement in the “Confidential to Editor” section, and submit your "Accept" recommendation.

Reviewer #5: (No Response)

2. Is the manuscript technically sound, and do the data support the conclusions?

Reviewer #5: Yes

3. Has the statistical analysis been performed appropriately and rigorously? 

Reviewer #5: Yes

4. Have the authors made all data underlying the findings in their manuscript fully available?

Reviewer #5: Yes

5. Is the manuscript presented in an intelligible fashion and written in standard English?

Reviewer #5: Yes

6. Review Comments to the Author

Reviewer #5: The authors have submitted a much improved manuscript. It is appreciated their extensive responsiveness to feedback. I have a few minor remaining pieces of feedback (see below). However, I believe this manuscript is much improved and an important contribution to the literature.

Results

1. For the network meta-analysis, section 3.4.2 a)—please label as “prescribed frequency” to ensure understanding this was prescribed vs. actual. It is appreciated this has been added to the discussion.

2. Similarly, please label “prescribed intensity-duration.”

3. You don’t necessarily have to repeat this phrase with additions to the methods data analytic section: “Under the premise of ensuring the random allocation of other variable factors apart from the…”

Discussion

1. I find the significant subgroup result of exercise performing better in prevention than postpartum to be quite interesting. If possible, it would enrich the discussion to include some thoughts as to why this might be important in terms of timing.

2. Line: 426-427: “Maternal” should say “mothers” And maternal could be quickly assisted by other peers in the event of an emergencies such as falls or other discomfort to ensure the safety of the exercise [46].

7. PLOS authors have the option to publish the peer review history of their article (what does this mean?). If published, this will include your full peer review and any attached files.

Reviewer #5: No

---

## [Author Response · Author response to Decision Letter 4]

23 Sep 2023

Dear Academic Editor and Reviewer,

Thanks for your comments of our manuscript entitled “Effectiveness of Aerobic Exercise in the Prevention and Treatment of Postpartum Depression: Meta-analysis and Network meta-analysis” [PONE-D-23-04024R4] - [EMID: aa83befb7a1a3e58]. Those comments are all valuable and helpful for revising and improving our paper. We have discussed all comments carefully and have made conscientious revision. Below, we respond to the main comments. We also have given further detail response in the up-loaded text-revised. 

Academic Editor (Jayonta Bhattacharjee)

Comment 1. Please review your reference list to ensure that it is complete and correct. If you have cited papers that have been retracted, please include the rationale for doing so in the manuscript text, or remove these references and replace them with relevant current references. Any changes to the reference list should be mentioned in the rebuttal letter that accompanies your revised manuscript. If you need to cite a retracted article, indicate the article’s retracted status in the References list and also include a citation and full reference for the retraction notice.

Response: We have carefully reviewed the reference list for our manuscript and have taken the necessary steps to ensure that it is complete and accurate. We have confirmed that none of the cited papers have been retracted, and therefore, there is no need to include a rationale for their inclusion in the manuscript text. Due to the expanded discussion section, two additional references have been added to the reference list. (Line 664-671 of the revised manuscript with track changes) We have not cited any retracted articles, so there is no need to indicate the retracted status in the References list or include retraction notices. We appreciate the thorough review process and are committed to maintaining the integrity of our references in accordance with the journal's guidelines. Thank you for your attention to this matter.

Reviewer 5

Thank you very much for your positive feedback on the revised manuscript and for your valuable suggestions. We will work to improve the manuscript according to your recommendations. Once again, we appreciate your patience and diligent work. Thank you for your assistance and support.

Results

Comment 1. For the network meta-analysis, section 3.4.2 a)—please label as “prescribed frequency” to ensure understanding this was prescribed vs. actual. It is appreciated this has been added to the discussion.

Response: The modifications have been made as requested. (Line 314-333 of the revised manuscript with track changes)

Comment 2. Similarly, please label “prescribed intensity-duration.”

Response: The modifications have been made as requested. (Line 335-356 of the revised manuscript with track changes)

Comment 3. You don’t necessarily have to repeat this phrase with additions to the methods data analytic section: “Under the premise of ensuring the random allocation of other variable factors apart from the…”

Response: The modifications have been made as requested. (Line 290-291,316-317,337-338 of the revised manuscript with track changes)

Discussion

Comment 1. I find the significant subgroup result of exercise performing better in prevention than postpartum to be quite interesting. If possible, it would enrich the discussion to include some thoughts as to why this might be important in terms of timing.

Response: The discussion on terms of timing has been added as requested. “Subgroup analysis suggests that the intervention objective (prevention vs. treatment) of exercise could potentially be a source of heterogeneity in this study (p=0.02＜0.05), indicating that the preventive effects of aerobic exercise are superior to the therapeutic effects. It is well known that engaging in appropriate aerobic exercise during pregnancy not only promotes pelvic mobility and increases birth canal space to alleviate maternal labor pain but also helps prevent pregnancy complications. There is evidence to suggest that postpartum depression (PPD) doesn’t exclusively occur after childbirth, as population-based studies indicate a similar 12% occurrence rate of depression during pregnancy. This suggests that PPD symptoms may originate during pregnancy in certain cases [43]. Furthermore, studies indicate that the antidepressant effects of exercise can persist for a period after exercise cessation [44]. Therefore, engaging in aerobic exercise during pregnancy may have a greater impact on preventing PPD compared to exercise as a treatment postpartum.” (Line 386-400 of the revised manuscript with track changes)

Comment 2. Line: 426-427: “Maternal” should say “mothers” And maternal could be quickly assisted by other peers in the event of an emergencies such as falls or other discomfort to ensure the safety of the exercise [46].

Response: The modifications have been made as requested. (Line 415 of the revised manuscript with track changes)

We firmly believe that these revisions further enhance the quality and readability of the manuscript. Once again, we extend our heartfelt thanks for your time and valuable input. Should you have any further questions or requests, please do not hesitate to contact us. We also modified some punctuation mark in the manuscript.

Best regards,

Hao Xu, Renyi Liu

---

## [Decision Letter · Decision Letter 5]

8 Oct 2023

Effectiveness of Aerobic Exercise in the Prevention and Treatment of Postpartum Depression: Meta-analysis and Network meta-analysis

PONE-D-23-04024R5

Dear Dr. Liu,

We’re pleased to inform you that your manuscript has been judged scientifically suitable for publication and will be formally accepted for publication once it meets all outstanding technical requirements.

Kind regards,

Jayonta Bhattacharjee

Academic Editor

PLOS ONE

Additional Editor Comments (optional):

Reviewers' comments:

Reviewer's Responses to Questions

**Comments to the Author**

1. If the authors have adequately addressed your comments raised in a previous round of review and you feel that this manuscript is now acceptable for publication, you may indicate that here to bypass the “Comments to the Author” section, enter your conflict of interest statement in the “Confidential to Editor” section, and submit your "Accept" recommendation.

Reviewer #5: All comments have been addressed

2. Is the manuscript technically sound, and do the data support the conclusions?

Reviewer #5: Yes

3. Has the statistical analysis been performed appropriately and rigorously? 

Reviewer #5: Yes

4. Have the authors made all data underlying the findings in their manuscript fully available?

Reviewer #5: (No Response)

5. Is the manuscript presented in an intelligible fashion and written in standard English?

Reviewer #5: Yes

6. Review Comments to the Author

Reviewer #5: (No Response)

7. PLOS authors have the option to publish the peer review history of their article (what does this mean?). If published, this will include your full peer review and any attached files.

Reviewer #5: No

---

## [Editor Report · Acceptance letter]

19 Jun 2023

PONE-D-23-04024R3 

Effectiveness of Aerobic Exercise in the Prevention and Treatment of Postpartum Depression: Meta-analysis and Network meta-analysis 

Dear Dr. Liu:

I'm pleased to inform you that your manuscript has been deemed suitable for publication in PLOS ONE. Congratulations! Your manuscript is now with our production department. 

Kind regards, 

on behalf of

Dr. Jayonta Bhattacharjee 

Academic Editor

PLOS ONE